# Multi-armed Bandits:
# Competing with Optimal Sequences

**Oren Anava**
The Voleon Group
Berkeley, CA
oren@voleon.com

**Zohar Karnin**
Yahoo! Research
New York, NY
zkarnin@yahoo-inc.com

## Abstract

We consider sequential decision making problem in the adversarial setting, where regret is measured with respect to the *optimal sequence of actions* and the feedback adheres the bandit setting. It is well-known that obtaining sublinear regret in this setting is impossible in general, which arises the question of *when can we do better than linear regret?* Previous works show that when the environment is guaranteed to vary slowly and furthermore we are given prior knowledge regarding its *variation* (i.e., a limit on the amount of changes suffered by the environment), then this task is feasible. The caveat however is that such prior knowledge is not likely to be available in practice, which causes the obtained regret bounds to be somewhat irrelevant.

Our main result is a regret guarantee that scales with the *variation* parameter of the environment, without requiring any prior knowledge about it whatsoever. By that, we also resolve an open problem posted by Gur, Zeevi and Besbes [8]. An important key component in our result is a statistical test for identifying non-stationarity in a sequence of independent random variables. This test either identifies non-stationarity or upper-bounds the absolute deviation of the corresponding sequence of mean values in terms of its total variation. This test is interesting on its own right and has the potential to be found useful in additional settings.

## 1 Introduction

Multi-Armed Bandit (MAB) problems have been studied extensively in the past, with two important special cases: the *Stochastic Multi-Armed Bandit*, and the *Adversarial (Non-Stochastic) Multi-Armed Bandit*. In both formulations, the problem can be viewed as a $T$-round repeated game between a player and nature. In each round, the player chooses one of $k$ actions[1] and observes the loss corresponding to this action only (the so-called *bandit feedback*). In the adversarial formulation, it is usually assumed that the losses are chosen by an all-powerful adversary that has full knowledge of our algorithm. In particular, the loss sequences need not comply with any distributional assumptions. On the other hand, in the stochastic formulation each action is associated with some mean value that does not change throughout the game. The feedback from choosing an action is an i.i.d. noisy observation of this action's mean value.

The performance of the player is traditionally measured using the *static regret*, which compares the total loss of the player with the total loss of the benchmark playing the best fixed action in hindsight. A stronger measure of the player's performance, sometimes referred to as *dynamic regret*[2] (or just *regret* for brevity), compares the total loss of the player with this of the optimal benchmark, playing the best possible *sequence* of actions. Notice that in the stochastic formulation both measures coincide, assuming that the benchmark has access to the parameters defining the random process of

the losses but not to the random bits generating the loss sequences. In the adversarial formulation this is clearly not the case, and it is not hard to show that attaining sublinear regret is impossible in general, whereas obtaining sublinear static regret is possible indeed. This can perhaps explain why most of the literature is concerned with optimizing the static regret rather than its dynamic counterpart.

Previous attempts to tackle the problem of regret minimization in the adversarial formulation mostly took advantage of some *niceness* parameter of nature (that is, some non-adversarial behavior of the loss sequences). This line of research becomes more and more popular, as full characterizations of the regret turn out to be feasible with respect to specific niceness parameters. In this work we focus on a broad family of such niceness parameters —usually called *variation* type parameters— originated from the work of [8] in the context of (dynamic) regret minimization. Essentially, we consider a MAB setting in which the mean value of each action can vary over time in an adversarial manner, and the feedback to the player is a noisy observation of that mean value. The variation is then defined as the sum of distances between the vectors of mean values over consecutive rounds, or formally,

$$\mathcal{V}_T \overset{\text{def}}{=} \sum_{t=2}^{T} \max_i |\mu_t(i) - \mu_{t-1}(i)|, \tag{1}$$

where $\mu_t(i)$ denotes the mean value of action $i$ at round $t$. Despite the presentation of $\mathcal{V}_T$ using the maximum norm, any other norm will lead to similar qualitative formulations. Previous approaches to the problem at hand relied on strong (and sometimes even unrealistic) assumptions on the variation (we refer the reader to Section 1.3, in which related work is discussed in detail). The natural question is whether it is possible to design an algorithm that does not require any assumptions on the variation, yet can achieve $o(T)$ regret whenever $\mathcal{V}_T = o(T)$. In this paper we answer this question in the affirmative and prove the following.

**Theorem** (Informal). *Consider a MAB setting with two arms and time horizon $T$. Assume that at each round $t \in \{1, \ldots, T\}$, the random variables of obtainable losses correspond to a vector of mean values $\mu_t$. Then, Algorithm 1 achieves a regret bound of $\tilde{O}\left(T^{0.771} + T^{0.82}\mathcal{V}_T^{0.18}\right)$.*

Our techniques rely on statistical tests designed to identify changes in the environment on the one hand, but exploit the best option observed so far in case there was no such significant environment change. We elaborate on the key ideas behind our techniques in Section 1.2.

## 1.1 Model and Motivation

A player is faced with a sequential decision making task: In each round $t \in \{1, \ldots, T\} = [T]$, the player chooses an action $i_t \in \{1, \ldots, k\} = [k]$ and observes loss $\ell_t(i_t) \in [0, 1]$. We assume that $\mathbb{E}[\ell_t(i)] = \mu_t(i)$ for any $i \in [k]$ and $t \in [T]$, where $\{\mu_t(i)\}_{t=1}^{T}$ are fixed beforehand by the adversary (that is, the adversary is *oblivious*). For simplicity, we assume that $\{\ell_t(i)\}_{t=1}^{T}$ are also generated beforehand. The goal of the player is to minimize the *regret*, which is henceforth defined as

$$\mathcal{R}_T = \sum_{t=1}^{T} \mu_t(i_t) - \sum_{t=1}^{T} \mu_t(i_t^*),$$

where $i_t^* = \arg\min_{i \in [k]}\{\mu_t(i)\}$. A sequence of actions $\{i_t\}_{t=1}^{T}$ has *no-regret* if $\mathcal{R}_T = o(T)$. It is well-known that generating no-regret sequences in our setting is generally impossible, unless the benchmark sequence is somehow limited (for example, in its total number of action switches) or alternatively, some characterization of $\{\mu_t(i)\}_{t=1}^{T}$ is given (in our case, $\{\mu_t(i)\}_{t=1}^{T}$ are characterized via the variation). While limiting the benchmark makes sense only when we have a strong reason to believe that an action sequence from the limited class has satisfactory performance, characterizing the environment is an approach that leads to guarantees of the following type:

> *If the environment is well-behaved (w.r.t. our characterization), then our performance is comparable with the optimal sequence of actions. If not, then no algorithm is capable of obtaining sublinear regret without further assumptions on the environment.*

Obtaining algorithms with such guarantee is an important task in many real-world applications. For example, an online forecaster must respond to time-related trends in her data, an investor seeks to detect trading trends as quickly as possible, a salesman should adjust himself to the constantly changing taste of his audience, and many other examples can be found. We believe that in many of

these examples the environment is often likely to change slowly, making guarantees of the type we present highly desirable.

## 1.2 Our Techniques

**An intermediate (noiseless) setting.** We begin with an easier setting, in which the observable losses are deterministic. That is, by choosing arm $i$ at round $t$, rather than observing the realization of a random variable with mean value $\mu_t(i)$ we simply observe $\mu_t(i)$. Note that $\{\mu_t\}_{t=1}^T$ are still assumed to be generated adversarially. In this setting, the following intuitive solution can be shown to work (for two arms): Pull each arm once and observe two values. Now, in each round pull the arm with the smaller loss w.p. $1 - o(1)$ and the other arm w.p. $o(1)$, where the latter is decreasing with time. As long as the mean values of the arms did not significantly shift compared to their original values, continue. Once a significant shift is observed, reset all counters and start over. We note that while the algorithm is simple, its analysis is not straightforward and contains some counterintuitive ideas. In particular, we show that the true (unknown) variation can be replaced with a crude proxy called the *observed variation* (to be later defined), while still maintaining mathematical tractability of the problem.

To see the importance of this proxy, let us first describe a different approach to the problem at hand that in particular extends directly the approach of [8] who show that if an upper-bound on $\mathcal{V}_T$ is known in advance, then the optimal regret is attainable. Therefore, one might guess that having an unbiased estimator for $\mathcal{V}_T$ will eliminate the need in this prior knowledge. Obtaining such an unbiased estimator is not hard (via importance sampling), but it turns out that it is not sufficient: the values of the variation to be identified are simply too small in order to be accurately estimated. Here comes into the picture the observed variation, which is loosely defined as the loss difference between two successive pulls of the same arm. Clearly, the true variation is only larger, but as we show, it cannot be much larger without us noticing it. We provide a complete analysis of the noiseless setting in Appendix A. This analysis is not directly used for dealing with the noisy setting but acts as a warm up and contains some of the key techniques used for it.

**Back to our (noisy) setting.** Here also we focus on the case of $k = 2$ arms. When the losses are stochastic the same basic ideas apply but several new major issues come up. In particular, here as well we present an algorithm that resets all counters and starts over once a significant change in the environment is detected. The similarity however ends here, mainly because of the noisy feedback that makes it hard to determine whether the changes we see are due to some environmental shift or due to the stochastic nature of the problem. The straightforward way of overcoming this is to forcefully divide the time into 'bins' in which we continuously pull the same arm. By doing this, and averaging the observed losses within a bin we can obtain feedback that is not as noisy. This meta-technique raises two major issues: The first is, how long should these bins be? A long period would eliminate the noise originating from the stochastic feedback but cripple our adaptive capabilities and make us more vulnerable to changes in the environment. The second issue is, if there was a change in the environment that is in some sense local to a single bin, how can we identify it? and assuming we did, when should we tolerate it?

The algorithm we present overcomes the first issue by starting with an *exploration phase*, where both arms are queried with equal probability. We advance to the next phase only once it is clear that the average loss of one arm is greater than the other, and furthermore, we have a solid estimate of the gap between them. In the next *exploitation phase*, we mimic the above algorithm for deterministic feedback by pulling the arms in bins of length proportional to the (inverted squared) gap between the arms. The techniques from above take care of the regret compared to a strategy that must be fixed inside the bins, or alternatively, against the optimal strategy if we were somehow guaranteed that there are no significant environment changes within bins. This leads us to the second issue, but first consider the following example.

**Example 1.** *During the exploration phase, we associated arm #1 with an expected loss of $0.5$ and arm #2 with an expected loss of $0.6$. Now, consider a bin in which we pull arm #1. In the first half of the bin the expected loss is $0.25$ and in the second it is $0.75$. The overall expected loss is $0.5$, hence without performing some test w.r.t. the pulls inside the bin we do not see any change in the environment, and as far as we know we mimicked the optimal strategy. The optimal strategy however can clearly do much better and we suffer a linear regret in this bin. Furthermore, the variation during the bin is constant! The good news is that in this scenario a simple test would determine that the outcome of the arm pulls inside the bin do not resemble those of an i.i.d. random variables, meaning that the environment change can be detected.*

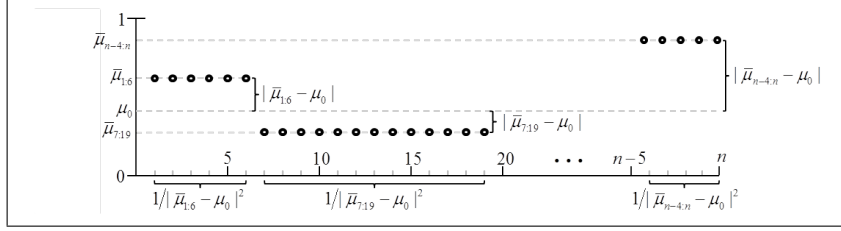

Figure 1: The optimal policy of an adversary that minimizes variation while maximizing deviation.

Example 1 clearly demonstrates the necessity of a statistical test inside a bin. However, there are cases in which the changes of the environment are unidentifiable and the regret suffered by any algorithm will be linear, as can be seen in the following example.

**Example 2.** *Assume that arm #1 has mean value of $0.5$ for all $t$, and arm #2 has mean value of $1$ with probability $0.5$ and $0$ otherwise. The feedback from pulling arm $i$ at a specific round is a Bernoulli random variable with the mean value of that arm. Clearly, there is no way to distinguish between these arms, and thus any algorithm would suffer linear regret. The point, however, is that the variation in this example is also linear, and thus linear regret is unavoidable in general.*

Example 2 shows that if the adversary is willing to put enough effort (in terms of variation), then linear regret is unavoidable. The intriguing question is whether the adversary can put some less effort (that is, to invest less than linear variation) and still cause us to suffer linear regret, while not providing us the ability to notice that the environment has changed. The crux of our analysis is the design of two tests, one per phase (exploration or exploitation), each is able to identify changes whenever it is possible or to ensure they do not hurt the regret too much whenever it is not. This building block, along with the 'outer bin regret' analysis mentioned above, allows us to achieve our result in this setting. The essence of our statistical tests is presented here, while formal statements and proofs are deferred to Section 2.

**Our statistical tests (informal presentation).** Let $X_1, \ldots, X_n \in [0, 1]$ be a sequence of realizations, such that each $X_i$ is generated from an arbitrary distribution with mean value $\mu_i$. Our task is to determine whether it is likely that $\mu_i = \mu_0$ for all $i$, where $\mu_0$ is a given constant. In case there is not enough evidence to reject this hypothesis, the test is required to bound the absolute deviation of $\mu^{\mathbf{n}} = \{\mu_i\}_{i=1}^n$ (henceforth denoted by $\|\mu^{\mathbf{n}}\|_{\text{ad}}$) in terms of its total variation[3] $\|\mu^{\mathbf{n}}\|_{\text{tv}}$. Assume for simplicity that $\bar{\mu}_{1:n} = \frac{1}{n}\sum_{i=1}^n \mu_i$ is close enough to $\mu_0$ (or even exactly equal to it), which eliminates the need to check the deviation of the average from $\mu_0$. We are thus left with checking the inner-sequence dynamics.

It is worthwhile to consider the problem from the adversary's point of view: The adversary has full control of the values of $\{\mu_i\}_{i=1}^n$, and his task is to deviate as much as he can from the average without providing us the ability to identify this deviation. Now, consider a partition of $[n]$ into consecutive segments, such that $(\mu_i - \mu_0)$ has the same sign for any $i$ within a segment. Given this partition, it can be shown that the optimal policy of an adversary that tries to minimize the total variation of $\{\mu_i\}_{i=1}^n$ while maximizing its absolute deviation, is to set $\mu_i$ to be equal within each segment. The length of a segment $[a, b]$ is thus limited to be at most $1/|\bar{\mu}_{a:b} - \mu_0|^2$, or otherwise the deviation is notable (this follows by standard concentration arguments). Figure 1 provides a visualization of this optimal policy. Summing the absolute deviation over the segments and using Hölder's inequality ensures that $\|\mu^{\mathbf{n}}\|_{\text{ad}} \leq n^{2/3}\|\mu^{\mathbf{n}}\|_{\text{tv}}^{1/3}$, or otherwise there exists a segment in which the distance between the realization average and $\mu_0$ is significantly large. Our test is thus the simple test that measures this distance for every segment. Notice that the test runs in polynomial time; further optimization might improve the polynomial degree, but is outside the scope of this paper.

The test presented above aims to bound the absolute deviation w.r.t. some given mean value. As such, it is appropriate only for the exploitation phase of our algorithm, in which a solid estimation of each arm's mean value is given. However, it turns out that bounding the absolute deviation with respect to some unknown mean value can be done using similar ideas, yet is slightly more complicated.

**Alternative approaches.** We point out that the approach of running a meta-bandit algorithm over (logarithmic many) instances of the algorithm proposed by [8] will be very difficult to pursue. In this approach, whenever an EXP3 instance is not chosen by the meta-bandit algorithm it is still forced to play an arm chosen by a different EXP3 instance. We are not aware of an analysis of EXP3 nor other algorithm equipped to handle such a harsh setting. Another idea that will be hard to pursue is tackling the problem using a doubling trick. This idea is common when parameters needed for the execution of an algorithm are unknown in advanced, but can in fact be guessed and updated if necessary. In our case, the variation is not observed due to the bandit feedback, and moreover, estimating it using importance sampling will lead to estimators that are too crude to allow a doubling trick.

## 1.3 Related Work

The question of whether (and when) it is possible to obtain bounds on other than the static regret is long studied in a variety of settings including *Online Convex Optimization* (OCO), *Bandit Convex Optimization* (BCO), *Prediction with Expert Advice*, and *Multi-Armed bandits* (MAB). Stronger notions of regret include the dynamic regret (see for instance [17, 4]), the adaptive regret [11], the strongly adaptive regret [5], and more. From now on, we focus on the dynamic regret only. Regardless of the setting considered, it is not hard to construct a loss sequence such that obtaining sublinear dynamic regret is impossible (in general). Thus, the problem of minimizing it is usually weakened in one of the two following forms: (1) restricting the benchmark; and (2) characterizing the niceness of the environment.

With respect to the first weakening form, [17] showed that in the OCO setting the dynamic regret can be bounded in terms of $C_T = \sum_{t=2}^{T} \|a_t - a_{t-1}\|$, where $\{a_t\}_{t=1}^{T}$ is the benchmark sequence. In particular, restricting the benchmark sequence with $C_T = 0$ gives the standard static regret result. [6] suggested that this type of result is attainable in the BCO setting as well, but we are not familiar with such result. In the MAB setting, [1] defined the *hardness* of a bencmark sequence as the number of its action switches, and bounded the dynamic regret in terms of this hardness. Here again, the standard static regret bound is obtained if the hardness is restricted to 0. The concept of bounding the dynamic regret in terms of the total number of action switches was studied by [14], in the setting of Prediction with Expert Advice.

With respect to the second weakening form, one can find an immense amount of MAB literature that uses stochastic assumptions to model the environment. In particular, [16] coined the term *restless bandits*; a model in which the loss sequences change in time according to an arbitrary, yet known in advance, stochastic process. To cope with the hard nature of this model, subsequent works offered approximations, relaxations, and more detailed models [3, 7, 15, 2]. Perhaps the first attempt to handle arbitrary loss sequences in the context of dynamic regret and MAB, appears in the work of [8]. In a setting identical to ours, the authors fully characterize the dynamic regret: $\Theta(T^{2/3}\mathcal{V}_T^{1/3})$, *if a bound on $\mathcal{V}_T$ is known in advance*. We provide a high-level description of their approach.

Roughly speaking, their algorithm divides the time horizon into (equally-sized) blocks and applies the EXP3 algorithm of [1] in each of them. This guarantees sublinear static regret w.r.t. the best fixed action in the block. Now, since the number of blocks is set to be much larger than the value of $\mathcal{V}_T$ (if $\mathcal{V}_T = o(T)$), it can be shown that in most blocks, the variation inside the block is $o(1)$ and the total loss of the best fixed action (within a block) turns out to be not very far from the total loss of the best sequence of actions. The size of the blocks (which is fixed and determined in advance as a function of $T$ and $\mathcal{V}_T$) is tuned accordingly to obtain the optimal rate in this case. The main shortcomings of this algorithms are the reliance on prior knowledge of $\mathcal{V}_T$, and the restarting procedure that does not take the variation into account.

We also note the work of [13], in which the two forms of weakening are combined together to obtain dynamic regret bounds that depend both on the complexity of the benchmark and on the niceness of the environment. Another line of work that is close to ours (at least in spirit) aims to minimize the static regret in terms of the variation (see for instance [9, 10]).

**A word about existing statistical tests.** There are actually many different statistical tests such as $z$-test, $t$-test, and more, that aim to determine whether a sample data comes from a distribution with a particular mean value. These tests however are not suitable for our setting since (1) they mostly require assumptions on the data generation (e.g., Gaussianity), and (2) they lack our desired bound on the total absolute deviation of the mean sequence in terms of its total variation. The latter is especially important in light of Example 2, which demonstrates that a mean sequence can deviate from its average without providing us any hint.

## 2 Competing with Optimal Sequences

Before presenting our algorithm and analysis we introduce some general notation and definitions. Let $\mathbf{X^n} = \{X_i\}_{i=1}^n \in [0, c]^n$ be a sequence of independent random variables, and denote $\mu_i = \mathbb{E}[X_i]$. For any $n_1, n_2 \in [n]$, where $n_1 \leq n_2$, we denote by $\bar{X}_{n_1:n_2}$ the average of $X_{n_1}, \ldots, X_{n_2}$, and by $\bar{X}_{n_1:n_2}^c$ the average of the other random variables. That is,

$$\bar{X}_{n_1:n_2} = \frac{1}{n_2 - n_1 + 1} \sum_{i=n_1}^{n_2} X_i \quad \text{and} \quad \bar{X}_{n_1:n_2}^c = \frac{1}{n - n_2 + n_1 - 1} \left( \sum_{i=1}^{n_1-1} X_i + \sum_{i=n_2+1}^{n} X_i \right).$$

We sometimes use the notation $\sum_{i \notin \{n_1, \ldots, n_2\}}$ for the second sum when $n$ is implied from the context. The expected values of $\bar{X}_{n_1:n_2}$ and $\bar{X}_{n_1:n_2}^c$ are denoted by $\bar{\mu}_{n_1:n_2}$ and $\bar{\mu}_{n_1:n_2}^c$, respectively. We use two additional quantities defined w.r.t. $n_1, n_2$:

$$\varepsilon_1(n_1, n_2) \stackrel{\text{def}}{=} \left( \frac{1}{n_2 - n_1 + 1} \right)^{1/2} \quad \text{and} \quad \varepsilon_2(n_1, n_2) \stackrel{\text{def}}{=} \left( \frac{1}{n_2 - n_1 + 1} + \frac{1}{n - n_2 + n_1 - 1} \right)^{1/2}.$$

We slightly abuse notation and define $\mathcal{V}_{n_1:n_2} \stackrel{\text{def}}{=} \sum_{i=n_1+1}^{n_2} |\mu_i - \mu_{i-1}|$ as the *total variation* of a mean sequence $\mu^{\mathbf{n}} = \{\mu_i\}_{i=1}^n \in [0, 1]^n$ over the interval $\{n_1, \ldots, n_2\}$.

**Definition 2.1.** *(weakly stationary, non-stationary) We say that $\mu^{\mathbf{n}} = \{\mu_i\}_{i=1}^n \in [0, 1]^n$ is $\alpha$-weakly stationary if $\mathcal{V}_{1:n} \leq \alpha$. We say that $\mu^{\mathbf{n}}$ is $\alpha$-non-stationary if it is not $\alpha$-weakly stationary*[4].

Throughout the paper, we mostly use these definitions with $\alpha = 1/\sqrt{n}$. In this case we will shorthand the notation and simply say that a sequence is weakly stationary (or non-stationary). In the sequel, we somewhat abuse notation and use capital letters $(X_1, \ldots, X_n)$ both for random variables and realizations. The specific use should be clear from the context, if not spelled out explicitly. Next, we define a notion of a concentrated sequence that depends on a parameter $T$. In what follows, $T$ will always be used as the time horizon.

**Definition 2.2.** *(concentrated, strongly concentrated) We say that a sequence $\mathbf{X^n} = \{X_i\}_{i=1}^n \in [0, c]^n$ is concentrated w.r.t. $\mu^{\mathbf{n}}$ if for any $n_1, n_2 \in [n]$ it holds that:*

*(1)* $\left| \bar{X}_{n_1:n_2} - \bar{\mu}_{n_1:n_2} \right| \leq \left( 2.5c^2 \log(T) \right)^{1/2} \varepsilon_1(n_1, n_2).$

*(2)* $\left| \bar{X}_{n_1:n_2} - \bar{X}_{n_1:n_2}^c - \bar{\mu}_{n_1:n_2} + \bar{\mu}_{n_1:n_2}^c \right| \leq \left( 2.5c^2 \log(T) \right)^{1/2} \varepsilon_2(n_1, n_2).$

*We further say that $\mathbf{X^n}$ is strongly concentrated w.r.t. $\mu^{\mathbf{n}}$ if any successive sub-sequence $\{X_i\}_{i=n_1}^{n_2} \subseteq \mathbf{X^n}$ is concentrated w.r.t. $\{\mu_i\}_{i=n_1}^{n_2}$.*

Whenever the mean sequence is inferred from the context, we will simply say that $\mathbf{X^n}$ is concentrated (or strongly concentrated). The parameters in the above definition are set so that standard concentration bounds lead to the statement that any sequence of independent random variables is strongly concentrated with high probability. The formal statement is given below and is proven in Appendix B.

**Claim 2.3.** *Let $\mathbf{X^T} = \{X_i\}_{i=1}^T \in [0, c]^T$ be a sequence of independent random variables, such that $T \geq 2$ and $c > 0$. Then, $\mathbf{X^T}$ is strongly concentrated with probability at least $1 - \frac{1}{T}$.*

### 2.1 Statistical Tests for Identifying Non-Stationarity

**TEST 1 (the offline test).** The goal of the offline test is to determine whether a sequence of realizations $\mathbf{X^n}$ is likely to be generated from a mean sequence $\mu^{\mathbf{n}}$ that is close (in a sense) to some given value $\mu_0$. This will later be used to determine whether a series of pulls of the same arm (inside a single *bin*) in the exploitation phase exhibit the same behavior as those observed in the exploration phase. We would like to have a two sided guarantee. If the means did not significantly shift the algorithm must state that the sequence is *weakly stationary*. On the other hand, if the algorithm states that the sequence is *weakly stationary* we require the absolute deviation of $\mu^{\mathbf{n}}$ to be bounded in terms of its total variation. We provide an analysis of Test 1 in Appendix B.

**Input:** a sequence $\mathbf{X^n} = \{X_i\}_{i=1}^n \in [0, c]^n$, and a constant $\mu_0 \in [0, 1]$.
**The test:** for any two indices $n_1, n_2 \in [n]$ such that $n_1 < n_2$, check whether

**Output:** *non-stationary* if such $n_1, n_2$ were found; *weakly stationary* otherwise.

TEST 1: (the offline test) The test aims to identify variation during the exploitation phase.

**Input:** a sequence $\mathbf{X^Q} = \{X_i\}_{i=1}^Q \in [0, c]^Q$, that is revealed gradually (one $X_i$ after the other).
**The test:** for $n = 2, 3, \ldots, Q$:
    (1) observe $X_n$ and set $\mathbf{X^n} = \{X_i\}_{i=1}^n$.
    (2) for any two indices $n_1, n_2 \in [n]$ such that $n_1 < n_2$, check whether

    and terminate the loop if such $n_1, n_2$ were found.
**Output:** *non-stationary* if the loop was terminated before $n = Q$, *weakly stationary* otherwise.

TEST 2: (the online test) The test aims to identify variation during the exploration phase.

**TEST 2 (the online test).** The online test gets a sequence $\mathbf{X^Q}$ in an online manner (one variable after the other), and has to stop whenever non-stationarity is exhibited (or the sequence ends). Here, the value of $Q$ is unknown to us beforehand, and might depend on the values of sequence elements $X_i$. The rationale here is the following: In the exploration phase of the main algorithm we sample the arms uniformly until discovering a significant gap between their average losses. While doing so, we would like to make sure that the regret is not large due to environment changes within the exploration process. We require a similar two sided guarantee as in the previous test, with an additional requirement informally ensuring that if we exit the block in the exploration phase the bound on the absolute deviation still applies. We provide the formal analysis in Appendix B.

## 2.2 Algorithm and Analysis

Having this set of testing tools, we proceed to provide a non-formal description of our algorithm. Basically, the algorithm divides the time horizon into blocks according to the variation it identifies. The blocks are denoted by $\{B_j\}_{j=1}^N$, where $N$ is the total number of blocks generated by the algorithm. The rounds within block $B_j$ are split into an exploration and exploitation phase, henceforth denoted $E_{j,1}$ and $E_{j,2}$ respectively. Each exploitation phase is further divided into bins, where the size of the bins within a block is determined in the exploration phase and does not change throughout the block. The bins within block $B_j$ are denoted by $\{A_{j,a}\}_{a=1}^{N_j}$, where $N_j$ is the total number of bins in block $B_j$. Note that both $N$ and $N_j$ are random variables. We use $t(j, \tau)$ to denote the $\tau$-th round in the exploration phase of block $B_j$, and $t(j, a, \tau)$ to denote the $\tau$-th round of the $a$-th bin in the exploitation phase of block $B_j$. As before, notice that $t(j, \tau)$ might vary from one run of the algorithm to another, yet is uniquely defined per one run of the algorithm (and the same holds for $t(j, a, \tau)$). Our algorithm is formally given in Algorithm 1, and the working scheme is visually presented in Figure 2. We discuss the extension of the algorithm to $k$ arms in Appendix D.

**Theorem 2.4.** *Set $\theta = \frac{1}{2}$ and $\lambda = \frac{\sqrt{37}-5}{2}$. Then, with probability of at least $1 - \frac{10}{T}$ the regret of Algorithm 1 is*

$$\mathcal{R}_T = \sum_{t=1}^T \mu_t(i_t) - \sum_{t=1}^T \mu_t(i_t^*) \leq O \left( \log(T) T^{0.82} \mathcal{V}_T^{0.18} + \log(T) T^{0.771} \right).$$

*Proof sketch.* Notice that the feedback we receive throughout the game is strongly concentrated with high probability, and thus it suffices to prove the theorem for this case. We analyze separately (a) blocks in which the algorithm did not reach the exploitation, and (b) blocks in which it did.

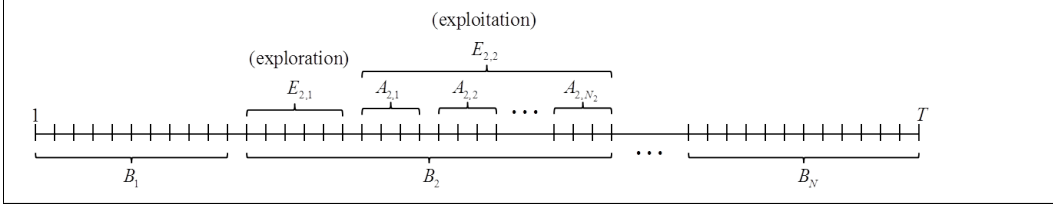

Figure 2: The time horizon is divided into blocks, where each block is split into an exploration phase and an exploitation phase. The exploitation phase is further divided into bins.

---

**Input:** parameters $\lambda$ and $\theta$.

**Algorithm:** In each block $j = 1, 2, \ldots$

**(Exploration phase)** In each round $\tau = 1, 2, \ldots$
  (1) Select action $i_{t(j,\tau)} \sim \mathrm{Uni}\{1, 2\}$ and observe loss $\ell_{t(j,\tau)}(i_{t(j,\tau)})$.
  (2) Set $X_{t(j,\tau)}(i) = \begin{cases} 2\ell_{t(j,\tau)}(i_{t(j,\tau)}) & \text{if } i = i_{t(j,\tau)} \\ 0 & \text{otherwise,} \end{cases}$
  and add $X_{t(j,\tau)}(i)$ (separately, for $i \in \{1, 2\}$) as an input to TEST 2.
  (3) If the test identifies non-stationarity (on either one of the actions), exit block. Otherwise, if

  $$\Delta \overset{\text{def}}{=} |\bar{X}_{t(j,1):t(j,\tau)}(1) - \bar{X}_{t(j,1):t(j,\tau)}(2)| \geq 16\big(\sqrt{10} + 2\big)^2 \log(T)\tau^{-\lambda/2}$$

  move to the next phase with $\hat{\mu}_0(i) = \bar{X}_{t(j,1):t(j,\tau)}(i)$ for $i \in \{1, 2\}$.

**(Exploitation phase)** Play in bins, each of size $n = 4/\Delta^2$. During each bin $a = 1, 2, \ldots$
  (1) Select action $i_{t(j,a,1)}, \ldots, i_{t(j,a,n)} = \begin{cases} \arg\min_i\{\hat{\mu}_0(i)\} & \text{w.p. } 1 - a^{-\theta} \\ \mathrm{Uni}\{1, 2\} & \text{otherwise,} \end{cases}$
  and observe losses $\{\ell_{t(j,a,\tau)}(i_{t(j,a,\tau)})\}_{\tau=1}^n$.
  (2) Run TEST 1 on $\{\ell_{t(j,a,\tau)}(i_{t(j,a,\tau)})\}_{\tau=1}^n$, and exit the block if it returned *non-stationary*.

Algorithm 1: An algorithm for the non-stationary multi-armed bandit problem.

**Analysis of part (a).** From TEST 2, we know that as long as the test does not identify non-stationarity in the exploration phase $E_1$, we can "trust" the feedback we observe as if we are in the stationary setting, i.e. standard stochastic MAB, up to an additive factor of $|E_1|^{2/3}\mathcal{V}_{E_1}^{1/3}$ to the regret. This argument holds even if TEST 2 identified non-stationarity, by simply excluding the last round. Now, since our stopping condition of the exploration phase is roughly $\Delta \geq \tau^{-\lambda/2}$, we suffer an additional regret of $|E_1|^{1-\lambda/2}$ throughout the exploration phase. This gives an overall bound of $|E_1|^{2/3}\mathcal{V}_{E_1}^{1/3} + |E_1|^{1-\lambda/2}$ for the regret (formally proven in Lemma C.4). The terms of the form $|E_1|^{1-\lambda/2}$ are problematic, as summing them may lead to an expression linear in $T$. To avoid this we use a lower bound on the variation $\mathcal{V}_{E_1}$ guaranteed by the fact that TEST 2 caused the block to end during the exploration phase. This lower bound allows to express $|E_1|^{1-\lambda/2}$ as $|E_1|^{1-\lambda/3}\mathcal{V}_{E_1}^{\lambda/3}$ leading to a meaningful regret bound on the entire time horizon (as detailed in Lemma C.5).

**Analysis of part (b).** The regret suffered in the exploration phase is bounded by the same arguments as before, where the bound on $|E_1|^{1-\lambda/2}$ is replaced by $|E_1|^{1-\lambda/2} \leq |B|^{1-\lambda/3}\mathcal{V}_B^{1-\lambda/3}$ with $B$ being the set of block rounds. This bound is achieved via a lower bound on $\mathcal{V}_B$, the variation in the block, guaranteed by the algorithm behavior along with fact that the block ended in the exploitation phase. For the regret in the exploitation phase, we first utilize the guarantees of TEST 1 to show that at the expense of an additive cost of $|E_2|^{2/3}\mathcal{V}_{E_2}^{1/3}$ to the regret, we may assume that there is no change to the environment inside bins. From hereon the analysis becomes very similar to that of the deterministic setting, as the noise corresponding to a bin is guaranteed to be lower than the gap $\Delta$ between the arms, and thus has no affect on the algorithm's performance. The final regret bound for blocks of type (b) comes from adding up the above mentioned bounds and is formally given in Lemma C.10. $\quad\square$

## Footnotes

[1]We sometimes use the terminology *arm* for an action throughout.

[2]The dynamic regret is occasionally referred to as *shifting regret* or *tracking regret* in the literature.

[3]We use standard notions of *total variation* and *absolute deviation*. That is, the total variation of a sequence $\{\mu_i\}_{i=1}^n$ is defined as $\|\mu^{\mathbf{n}}\|_{\text{tv}} = \sum_{i=2}^n |\mu_i - \mu_{i-1}|$, and its absolute deviation is $\|\mu^{\mathbf{n}}\|_{\text{ad}} = \sum_{i=1}^n |\mu_i - \bar{\mu}_{1:n}|$, where $\bar{\mu}_{1:n} = \frac{1}{n}\sum_{i=1}^n \mu_i$.

[4]We use stationarity-related terms to classify mean sequences. Our definition might be not consistent with stationarity-related definitions in the statistical literature, which are usually used to classify sequences of random variables based on higher moments or CDF's.

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
