[Supplementary Material]

# Supplementary Material
# Multi-armed Bandits: Competing with Optimal Sequences

## A  Analysis of the Deterministic Feedback Setting

In this section we present an intermediate setting, in which $\ell_t(i) = \mu_t(i)$ for any $t \in [T]$ and $i \in [k]$ (that is, the feedback is noiseless). This setting demonstrates some of the techniques later used in the more complex stochastic feedback scenario. Our solution is given in Algorithm 2.

---

Let $t(j, \tau)$ denote the time index of the $\tau$-th round in the $j$-th block.

**Initialize:** $\mathcal{OV}_{B_j} = 0$ for $j \geq 1$.

**Algorithm:** In each block $j = 1, 2, \ldots$

**(Exploration phase)**

    (1) Select actions $1, \ldots, k$.

    (2) Define the corresponding losses as the $k$-dimensional vector $\ell_0$.

**(Exploitation phase)**    Set $\tau = k + 1$. While $\tau \leq \frac{k}{4\mathcal{OV}_{B_j}^2}$ do

    (1) Select action $i_{t(j,\tau)} = \begin{cases} \arg\min_i \{\ell_0(i)\} & \text{w.p. } 1 - (k/\tau)^{1/2}, \\ \text{Uni}\{1, \ldots, k\} & \text{otherwise.} \end{cases}$

    (2) Update $\mathcal{OV}_{B_j} = |\ell_{t(j,\tau)}(i_{t(j,\tau)}) - \ell_0(i_{t(j,\tau)})|$ and increment $\tau$.

---

Algorithm 2: An algorithm for the intermediate setting, in which the feedback is noiseless.

**Theorem A.1.** *With probability at least* $1 - \frac{2}{T}$ *the regret of Algorithm 2 is bounded by*

$$\mathcal{R}_T = \sum_{t=1}^{T} \mu_t(i_t) - \mu_t(i_t^*) \leq 10\log(T)k^{1/2}T^{1/2} + 20\log(T)k^{1/3}T^{2/3}\mathcal{V}_T^{1/3}.$$

*Proof.* In what follows we provide a high probability regret bound stating that given some event $\mathcal{A}$, that occurs w.h.p. over the internal randomness of our algorithm, the regret is bounded. Our analysis applies for every block separately hence the high level structure of our proof is as follows. We fix some initial time point $t_{\text{start}}$ and consider the algorithm's performance on a block starting at $t_{\text{start}}$. We show that w.h.p. the regret of the algorithm is bounded in that block hence via union bound over all possible values of $t_{\text{start}}$ we obtain a high probability bound in each block.

We start with some definitions in order to characterize the random bits used by our algorithm. For any $t_{\text{start}} \in [T]$ we define an sequence $i_{k+1}^{(t_{\text{start}})}, i_{k+2}^{(t_{\text{start}})}, \ldots,$ of random elements in $[k]$ such that $i_t^{(t_{\text{start}})}$ is equal to 1 w.p. $1 - (k/t)^{1/2}$ and uniform in $[k]$ otherwise. Notice that these random numbers can completely characterize the behavior of the algorithm; For a block $B_j$ starting at time $t_{\text{start}}$, the arms are chosen according to the sequence of numbers corresponding to the time stamp $t_{\text{start}}$. Specifically, the arms are assigned numbers according to their rank in the exploration phase, performed at times $t_{\text{start}}$ through $t_{\text{start}} + k$ and during the exploration phase, the chosen arm at time $t > t_{\text{start}} + k$ is the one corresponding to the chosen random number.

We continue to define the mentioned event $\mathcal{A}$ w.r.t. a single block $B$. Assume first w.l.o.g., for ease of notations that $t_{\text{start}} = 1$, and that during the exploration phase, the ranking of the arms via their losses exactly matches their indices. We also write $i_t$ in short for $i_t^{(1)}$ according to the above definition of the random sequence. The event we are interested in is the intersection of two sub-events. The first informally states that the best performing arm in the exploration phase is chosen in all but a small fraction of the time during the exploitation phase of a block. Formally, event $\mathcal{A}_1$ occurs if for all $t_{\text{end}}$ (denoting a possible value for the end of the block) we have that

$$|\{k < t \leq t_{\text{end}} | i_t \neq 1\}| \leq \sqrt{11\log(T)k t_{\text{end}}} \tag{2}$$

**Lemma A.2.** *Event $\mathcal{A}_1$ occurs with probability at least $1 - \frac{1}{T^2}$.*

*Proof.* The claim is a direct application of Hoeffding's inequality (given in Theorem B.1). Using the notation there, we define $I_t$ as the indicator of whether $i_{t+k} \neq 1$, and $n = t_{\text{end}} - k$. The expected value of $I_t$ is $(k/(t+k))^{1/2}$ hence

$$\mathbb{E}\left[\sum_{t=1}^{n} I_t\right] = \sum_{t=1}^{n}(k/(t+k))^{1/2} \leq 2\sqrt{k(n+k+1)}$$

Setting $\varepsilon = \frac{\sqrt{1.5\log(T)k(n+k+1)}}{n}$ gives

$$\Pr\left(\sum_{t=1}^{n} I_t > 2\sqrt{k(n+k+1)} + \sqrt{1.5\log(T)k(n+k+1)}\right) \leq e^{-2n\varepsilon^2} \leq 1/T^3$$

where the last inequality holds since $2n\varepsilon^2 = 3\log(T)\sqrt{\frac{k(n+k+1)}{n}} \geq 3\log(T)$. Restated in terms of the claim we obtain

$$\Pr\left(|\{k < t \leq t_{\text{end}}|i_t \neq 1\}| > \sqrt{11\log(T)kt_{\text{end}}}\right) \leq 1/T^3$$

and the required result follows via a union bound over the $T$ possible values of $t_{\text{end}}$. $\square$

The next event informally states that the overall regret of a policy choosing the arm that performed best during the exploration phase has a small regret. We denote by $i^*$ the best action in the exploration phase of the block (that is w.l.o.g. assumed to be 1), that is,

$$i^* = \arg\min_{\tau \in [k]}\{\ell_\tau(\tau)\},$$

For $t > k$, representing a possible time index of the exploitation phase of the block, define $\Delta_t = \ell_t(i^*) - \ell_t(i_t^*)$, where $i_t^*$ is the best action at time $t$, and $\delta_t = \mathbf{1}\{\Delta_t \geq (k/t)^{1/2}\}$. Additionally denote by $\{t_n\}_{n=1}^{\infty}$ the rounds in which $\delta_{t_n} = 1$, sorted according to the cardinality of $t_n$, and define

$$s_{\min} = \arg\min_s\left\{\sum_{n=1}^{s}\frac{1}{\sqrt{k(t_n)}} \geq 2\log(T)\right\}. \tag{3}$$

If no such $s$ exists, define $s_{\min} = T$. Let $Y$ denote the total number of rounds in the block for which $\delta_t = 1$, that is, $Y = \sum_{t \in B} \delta_t$. Event $\mathcal{A}_2$ is the event in which $Y \leq s_{\min}$. In Lemma A.4 below we prove that this occurs w.p. at least $1 - \frac{1}{T^2}$.

Now that the events are defined we prove the regret bound conditioned on them. For a block $B$ starting at some fixed time (w.l.o.g. 1), the regret endured during the block can be written as follows

$$\mathcal{R}_B = \sum_{t=1}^{|B|}(\ell_t(i_t) - \ell_t(i_t^*)) = \underbrace{\sum_{t=1}^{|B|}(\ell_t(i_t) - \ell_t(i^*))}_{\mathcal{R}_B^{(1)}} + \underbrace{\sum_{t=1}^{|B|}(\ell_t(i^*) - \ell_t(i_t^*))}_{\mathcal{R}_B^{(2)}}. \tag{4}$$

The occurrence of event $\mathcal{A}_1$, along with the fact that the losses are in $[0,1]$, guarantees that regardless of the value of $|B|$ it holds that

$$\mathcal{R}_B^{(1)} = \sum_{t=1}^{|B|}(\ell_t(i_t) - \ell_t(i^*)) \leq k + |\{k < t \leq |B||i_t \neq 1\}| \leq k + \sqrt{11\log(T)k|B|}$$

The occurrence of event $\mathcal{A}_2$ guarantees that for the variable $Y$ described above

$$\frac{Y-1}{\sqrt{kt_Y}} \leq \sum_{n=1}^{Y-1}\frac{1}{\sqrt{kt_n}} < 2\log(T),$$

or equivalently[5], $Y < 1 + 2\log(T)\sqrt{kt_Y} \le 3\log(T)k^{1/2}|B|^{1/2}$. It follows that

$$\mathcal{R}_B^{(2)} = \sum_{t \in B} \ell_t(i^*) - \ell_t(i_t^*) = \sum_{t \in B; \delta_t = 1} \ell_t(i^*) - \ell_t(i_t^*) + \sum_{t \in B; \delta_t \ne 1} \ell_t(i^*) - \ell_t(i_t^*)$$

$$\le Y + \sum_{t=1}^{|B|} \left(\frac{k}{t}\right)^{1/2}$$

$$\le 3\log(T)k^{1/2}|B|^{1/2} + 2k^{1/2}|B|^{1/2}$$

$$\le 5\log(T)k^{1/2}|B|^{1/2}.$$

Combining the bounds on the regret types lead to $\mathcal{R}_B \le 10\log(T)k^{1/2}|B|^{1/2}$. Now, by taking a union bound over all possible $T$ starting points for a block we get that w.p. at least $1 - \frac{2}{T}$, the total regret is bounded by the following expression, where $N$ denotes the overall number of blocks

$$\mathcal{R}_T \le 10\log(T)\sum_{j=1}^{N} k^{1/2}|B_j|^{1/2}. \tag{5}$$

Finally, we use Hölder's inequality and the guaranteed bound on the variation in each block to achieve the final regret bound

$$\sum_{j=1}^{N} k^{1/2}|B_j|^{1/2} = k^{1/2}|B_N|^{1/2} + k^{1/3}\sum_{j=1}^{N-1} |B_j|^{2/3}(k/|B_j|)^{1/6}$$

$$\le k^{1/2}T^{1/2} + k^{1/3}\left(\sum_{j=1}^{N-1} |B_j|\right)^{2/3}\left(\sum_{j=1}^{N-1} (k/|B_j|)^{1/2}\right)^{1/3}$$

$$\overset{(a)}{\le} k^{1/2}T^{1/2} + k^{1/3}T^{2/3}\left(2\sum_{j=1}^{N-1} \mathcal{OV}_{B_j}\right)^{1/3}$$

$$\overset{(b)}{\le} k^{1/2}T^{1/2} + 2k^{1/3}T^{2/3}\mathcal{V}_T^{1/3},$$

where (a) follows by the stopping condition of the exploitation phase; and (b) holds since the observed variation is only less or equal the true variation. Specifically, for any $\tau > k$ and block $j$ with over $\tau$ rounds, it holds that

$$\mathcal{OV}_{B_j} = |\ell_{t(j,\tau)}(i_t) - \ell_{t(j,i_t)}(i_t)| \le \sum_{s=i_t+1}^{\tau} |\ell_{t(j,s)}(i) - \ell_{t(j,s)-1}(i)| \le \mathcal{V}_{B_j}.$$

The reason for separately dealing with the last block is because this block has no lower bound guarantee on the observed variation since it is terminated due to the end of the time horizon. Substituting the above in Eq. (5) gives the result stated in the theorem.

$\square$

We turn to prove the technical lemma bounding $Y_j$ in the above theorem. To this end, we begin with an auxiliary lemma analyzing the probability of stopping a block in a round in which $\delta_{t(j,\tau)} = 1$.

**Lemma A.3.** *Consider a block $B_j$ and let $t(\tau, j) \in B_j$ be such that $\delta_{t(j,\tau)} = 1$. Then, the probability that Algorithm 2 stops at round $t$ (given that it did not stop before) is at least $1/\sqrt{k\tau}$.*

*Proof.* Using our notation, we know that $\Delta_t = \ell_t(i_{B_j}^*) - \ell_t(i_t^*) \ge (k/\tau)^{1/2}$ for the $t$ specified in the lemma. Denote $\ell_0(i) = \ell_{t(j,i)}(i)$, that is, the value assigned to action $i$ in the exploration phase of block $B_j$. Thus, we can write

$$\ell_t(i_{B_j}^*) - \ell_0(i_{B_j}^*) + \ell_0(i_t^*) - \ell_t(i_t^*) + \ell_0(i_{B_j}^*) - \ell_0(i_t^*) \ge (k/\tau)^{1/2},$$

where $\ell_0(i^*_{B_j}) - \ell_0(i^*_t) \leq 0$ from the definition. This implies that either

$$(a) \ \ell_t(i^*_{B_j}) - \ell_0(i^*_{B_j}) \geq \frac{1}{2}(k/\tau)^{1/2} \quad \text{or} \quad (b) \ \ell_0(i^*_t) - \ell_t(i^*_t) \geq \frac{1}{2}(k/\tau)^{1/2}.$$

Recall that Algorithm 2 stops whenever it encounters an action $i$ such that $|\ell_t(i) - \ell_0(i)| \geq \frac{1}{2}(k/\tau)^{1/2}$. Thus, if (a) holds then Algorithm 2 stops with probability $1 - (k/\tau)^{1/2} + (k/\tau)^{1/2}(1/k)$, which is the probability of choosing action $i^*_{B_j}$. If (b) holds, then Algorithm 2 stops with probability $(k/\tau)^{1/2}(1/k) = 1/\sqrt{k\tau}$, which is the probability of choosing action $i^*_t$. We therefore get that the probability of stopping at round $t$ is at least $1/\sqrt{k\tau}$, as required. $\square$

**Lemma A.4.** *Set $j$ and let $s_{min}$ and $\tau_{s_{min}}$ be as defined in Eq. (3) for block $B_j$. Then, it holds that*

$$\mathbb{P}\left(|B_j| \leq \tau_{s_{min}}\right) \geq 1 - \frac{1}{T^2}.$$

*Proof.* We bound $\log\left(\mathbb{P}\left(|B_j| > \tau_{s_{min}}\right)\right)$ from above and the result will follow. Here and on the sequel, log refers to the natural logarithm. Thus,

$$\log\left(\mathbb{P}\left(|B_j| > \tau_{s_{min}}\right)\right) \leq \log\left(\prod_{n=1}^{s_{min}}\left(1 - \frac{1}{\sqrt{k\tau_n}}\right)\right)$$

$$= \sum_{n=1}^{s_{min}} \log\left(1 - \frac{1}{\sqrt{k\tau_n}}\right) \leq \sum_{n=1}^{s_{min}} -\frac{1}{\sqrt{k\tau_n}} \leq -2\log(T),$$

where we use the fact that when $\delta_{t(j,\tau)} = 1$, then the probability of stopping is at least $1/\sqrt{k\tau}$ as shown in Lemma A.3. The rounds in which $\delta_{t(j,\tau)} = 0$ can simply be ignored for this calculation, as the algorithm cannot stop at these rounds. It follows that $\mathbb{P}\left(|B_j| \leq \tau_{s_{min}}\right) \geq 1 - \frac{1}{T^2}$. $\square$

# B  Analysis of the Statistical Tests

*Proof of Claim 2.3.* The lemma is an immediate corollary of the following two versions of Hoeffding's inequality [12], denoted Hoeffding-1 and Hoeffding-2.

**Theorem B.1** (Hoeffding-1). *Let $\mathbf{X^n} = \{X_i\}_{i=1}^n \in [0, c]^n$ be a sequence of independent random variables, such that $\mathbb{E}[X_i] = \mu_i$. Then, it holds that*

$$\mathbb{P}\left(\left|\bar{X}_{1:n} - \bar{\mu}_{1:n}\right| \geq \varepsilon\right) \leq 2e^{-\frac{2n\varepsilon^2}{c^2}}.$$

**Corollary** (Hoeffding-2). *Let $\mathbf{X^{n+m}} = \{X_i\}_{i=1}^{n+m} \in [0, c]^{n+m}$ be a sequence of independent random variables, such that $\mathbb{E}[X_i] = \mu_i$. Then, it holds that*

$$\mathbb{P}\left(\left|\bar{X}_{1:n} - \bar{X}_{n+1:n+m} - (\bar{\mu}_{1:n} - \bar{\mu}_{n+1:n+m})\right| \geq \varepsilon\right) \leq 2e^{-\frac{2\varepsilon^2}{(n^{-1}+m^{-1})c^2}}.$$

$\square$

**Claim B.2.** *Let $\mathbf{X^n} = \{X_i\}_{i=1}^n \in [0, c]^n$ be a sequence generated from $\mu^{\mathbf{n}}$, and assume that $\mathbf{X^n}$ is concentrated. Then, by executing TEST 1 on $\mathbf{X^n}$ and $\mu_0 \in [0, 1]$ we know that:*

*(1) if $\mu^{\mathbf{n}}$ is weakly stationary and $|\bar{\mu}_{1:n} - \mu_0| \leq n^{-1/2}\log^{1/2}(T)$, then the test is never wrong (that is, it always classifies $\mu^{\mathbf{n}}$ as weakly stationary).*

*(2) if it classifies $\mu^{\mathbf{n}}$ as weakly stationary and $\mathcal{V}_{1:n} \geq \left(\sqrt{10}c + 2\right)\log^{1/2}(T)n^{-1/2}$, then*

$$\sum_{i=1}^n |\mu_i - \mu_0| \leq \left(\sqrt{10}c + 2\right)^{2/3}\log^{1/3}(T)n^{2/3}\mathcal{V}_{1:n}^{1/3}.$$

*(3) if it classifies $\mu^{\mathbf{n}}$ as weakly stationary and $\mathcal{V}_{1:n} \leq \left(\sqrt{10}c + 2\right)\log^{1/2}(T)n^{-1/2}$, then for any $i \in [n]$ it holds that*

$$|\mu_i - \mu_0| \leq 2\left(\sqrt{10}c + 2\right)\log^{1/2}(T)n^{-1/2}.$$

*Proof.* To prove (1), assume that $\mathbf{X^n}$ is concentrated and $\mu^{\mathbf{n}}$ is weakly stationary. Thus, for any $n_1, n_2 \in [n]$ it holds that

$$
\begin{aligned}
\left|\bar{X}_{n_1:n_2} - \mu_0\right| &= \left|\bar{X}_{n_1:n_2} - \bar{\mu}_{n_1:n_2} + \bar{\mu}_{n_1:n_2} - \bar{\mu}_{1:n} + \bar{\mu}_{1:n} - \mu_0\right| \\
&\overset{(a)}{\leq} \left|\bar{X}_{n_1:n_2} - \bar{\mu}_{n_1:n_2}\right| + \left|\bar{\mu}_{n_1:n_2} - \bar{\mu}_{1:n}\right| + \left|\bar{\mu}_{1:n} - \mu_0\right| \\
&\overset{(b)}{\leq} \left(2.5c^2 \log(T)\right)^{1/2} \varepsilon_1\left(n_1, n_2\right) + n^{-1/2} + n^{-1/2} \log^{1/2}(T) \\
&\leq \left(\sqrt{2.5}c + 2\right) \log^{1/2}(T) \varepsilon_1\left(n_1, n_2\right),
\end{aligned}
$$

where (a) follows by the triangle inequality; and (b) holds since $\mathbf{X^n}$ is concentrated and $\mu^{\mathbf{n}}$ is weakly stationary. It follows that TEST 1 classifies $\mathbf{X^n}$ as weakly stationary.

To prove (2), assume that TEST 1 classified $\mu^{\mathbf{n}}$ as weakly stationary. Thus, for any $n_1, n_2 \in [n]$ it holds that

$$
\left|\bar{X}_{n_1:n_2} - \mu_0\right| \leq \left(\sqrt{2.5}c + 2\right) \log^{1/2}(T) \varepsilon_1\left(n_1, n_2\right)
$$

Let $n_1, \ldots, n_m$ be rounds for which $\{\mu_{n_i} \geq \mu_0 \text{ and } \mu_{n_i+1} < \mu_0\}$ or $\{\mu_{n_i} < \mu_0 \text{ and } \mu_{n_i+1} \geq \mu_0\}$. We deal with the case where $m = 0$ later. Additionally, denote $n_0 = 0$ and $n_{m+1} = n$. Now, the term we are interested in bounding can be written equivalently as follows

$$
\begin{aligned}
\sum_{i=1}^{n} |\mu_i - \mu_0| &= \sum_{i=0}^{m} (n_{i+1} - n_i) \left|\bar{\mu}_{n_i+1:n_{i+1}} - \mu_0\right| \\
&= \sum_{i=0}^{m} (n_{i+1} - n_i) \left|\bar{\mu}_{n_i+1:n_{i+1}} - \mu_0\right|^{2/3} \left|\bar{\mu}_{n_i+1:n_{i+1}} - \mu_0\right|^{1/3}. \quad (6)
\end{aligned}
$$

Next, we bound the terms of the form $\left|\bar{\mu}_{n_i+1:n_{i+1}} - \mu_0\right|$.

$$
\begin{aligned}
\left|\bar{\mu}_{n_i+1:n_{i+1}} - \mu_0\right| &\leq \left|\bar{\mu}_{n_i+1:n_{i+1}} - \bar{X}_{n_i+1:n_{i+1}}\right| + \left|\bar{X}_{n_i+1:n_{i+1}} - \mu_0\right| \\
&\leq \sqrt{2.5}c \log^{1/2}(T) \varepsilon_1(n_i + 1, n_{i+1}) + \left(\sqrt{2.5}c + 2\right) \log^{1/2}(T) \varepsilon_1(n_i + 1, n_{i+1}) \\
&= \left(\sqrt{10}c + 2\right) \log^{1/2}(T) \varepsilon_1(n_i + 1, n_{i+1}).
\end{aligned}
$$

By substituting this in Eq. (6) we get that

$$
\begin{aligned}
\sum_{i=1}^{n} |\mu_i - \mu_0| &\leq \sum_{i=0}^{m} (n_{i+1} - n_i) (\sqrt{10}c + 2)^{2/3} \log^{1/3}(T) (n_{i+1} - n_i)^{-1/3} \left|\bar{\mu}_{n_i+1:n_{i+1}} - \mu_0\right|^{1/3} \\
&= \left(\sqrt{10}c + 2\right)^{2/3} \log^{1/3}(T) \sum_{i=0}^{m} (n_{i+1} - n_i)^{2/3} \left|\bar{\mu}_{n_i+1:n_{i+1}} - \mu_0\right|^{1/3} \\
&\overset{(a)}{\leq} \left(\sqrt{10}c + 2\right)^{2/3} \log^{1/3}(T) \left(\sum_{i=0}^{m} (n_{i+1} - n_i)\right)^{2/3} \left(\sum_{i=0}^{m} \left|\bar{\mu}_{n_i+1:n_{i+1}} - \mu_0\right|\right)^{1/3} \\
&\leq \left(\sqrt{10}c + 2\right)^{2/3} \log^{1/3}(T) \left(\sum_{i=0}^{m} (n_{i+1} - n_i)\right)^{2/3} \left(\sum_{i=1}^{m} \left|\bar{\mu}_{n_i+1:n_{i+1}} - \bar{\mu}_{n_{i-1}+1:n_i}\right|\right)^{1/3} \\
&\overset{(b)}{\leq} \left(\sqrt{10}c + 2\right)^{2/3} \log^{1/3}(T) \left(\sum_{i=0}^{m} (n_{i+1} - n_i)\right)^{2/3} \left(\sum_{i=1}^{m} \mathcal{V}_{n_{i-1}+1:n_{i+1}}\right)^{1/3} \\
&\overset{(c)}{\leq} \left(\sqrt{10}c + 2\right)^{2/3} \log^{1/3}(T) n^{2/3} \mathcal{V}_{1:n}^{1/3},
\end{aligned}
$$

where (a) follows from Hölder's inequality; (b) has a technical proof given in Claim B.3 below; and (c) follows from the fact that $\sum_{i=1}^{m} \mathcal{V}_{n_{i-1}+1:n_{i+1}} \leq \mathcal{V}_{1:n}$.

We deal now with the case of $m = 0$ (that is, $\mu_0 \notin [\min_i\{\mu_i\}, \max_i\{\mu_i\}]$). In this case, we can bound the term of interest as follows:

$$\sum_{i=1}^{n} |\mu_i - \mu_0| = |\bar{\mu}_{1:n} - \mu_0| \cdot n \leq \left( |\bar{\mu}_{1:n} - \bar{X}_{1:n}| + |\bar{X}_{1:n} - \mu_0| \right) \cdot n$$

$$\leq \left( \sqrt{2.5}c \log^{1/2}(T)\varepsilon_1(1, n) + \left(\sqrt{2.5}c + 2\right) \log^{1/2}(T)\varepsilon_1(1, n) \right) \cdot n$$

$$\leq \left( \sqrt{10}c + 2 \right) \log^{1/2}(T)n^{1/2}.$$

Now, since we assumed that $\mathcal{V}_{1:n} \geq \left(\sqrt{10}c + 2\right)\log^{1/2}(T)n^{-1/2}$, then we can replace the bound above with

$$\sum_{i=1}^{n} |\mu_i - \mu_0| \leq \left(\sqrt{10}c + 2\right)^{2/3} \log^{1/3}(T)n^{2/3}\mathcal{V}_{1:n}^{1/3},$$

which proves the statement of the claim for the case of $m = 0$ as well.

To prove (3), assume that (a) $\mathcal{V}_{1:n} \leq \left(\sqrt{10}c + 2\right)\log^{1/2}(T)n^{-1/2}$; (b) that the sequence is concentrated; and (c) that the test classified the sequence as weakly stationary. Repeating the same calculation from before gives

$$|\mu_i - \mu_0| \leq |\mu_i - \bar{\mu}_{1:n}| + |\bar{\mu}_{1:n} - \bar{X}_{1:n}| + |\bar{X}_{1:n} - \mu_0|$$

$$\leq \left(\sqrt{10}c + 2\right)\log^{1/2}(T)n^{-1/2} + \sqrt{2.5}c \log^{1/2}(T)n^{-1/2} + \left(\sqrt{2.5}c + 2\right)\log^{1/2}(T)n^{-1/2}$$

$$\leq 2\left(\sqrt{10}c + 2\right)\log^{1/2}(T)n^{-1/2},$$

for any $i \in [n]$, as required. $\square$

**Claim B.3.** *Let $\mu_1, \ldots, \mu_n$ be a sequence of scalars, and let $m < n$. Assume that $\{\mu_1, \ldots, \mu_m \geq 0$ and $\mu_{m+1}, \ldots, \mu_n \leq 0\}$ or $\{\mu_1, \ldots, \mu_m \leq 0$ and $\mu_{m+1}, \ldots, \mu_n \geq 0\}$. Then,*

$$\left| \frac{1}{m} \sum_{i=1}^{m} \mu_i - \frac{1}{n-m} \sum_{i=m+1}^{n} \mu_i \right| \leq \sum_{i=1}^{n-1} |\mu_{i+1} - \mu_i|.$$

*Proof.* Assume without loss of generality that $\{\mu_1, \ldots, \mu_m \leq 0$ and $\mu_{m+1}, \ldots, \mu_n \geq 0\}$, and let $i_1 = \arg\min_{i \in \{1, \ldots, m\}}\{\mu_i\}$ and $i_2 = \arg\max_{i \in \{m+1, \ldots, n\}}\{\mu_i\}$. Thus, we have

$$\left| \frac{1}{m} \sum_{i=1}^{m} \mu_i - \frac{1}{n-m} \sum_{i=m+1}^{n} \mu_i \right| \leq \mu_{i_2} - \mu_{i_1} = \sum_{i=i_1}^{i_2-1} \mu_{i+1} - \mu_i \leq$$

$$\sum_{i=i_1}^{i_2-1} |\mu_{i+1} - \mu_i| \leq \sum_{i=1}^{n-1} |\mu_{i+1} - \mu_i|.$$

$\square$

**Claim B.4.** *Let $\mathbf{X}^{\mathbf{Q}} = \{X_i\}_{i=1}^{Q} \in [0, c]^Q$ be a sequence generated from $\mu^{\mathbf{Q}}$ and revealed to us gradually (one realization after the other). Assume further that $\mathbf{X}^{\mathbf{Q}}$ is strongly concentrated. Then, by executing TEST 2 on $\mathbf{X}^{\mathbf{Q}}$ we know that:*

*(1) if $\mu^{\mathbf{Q}}$ is weakly stationary, then the test classifies it as weakly stationary.*

*(2) if it classifies $\mu^{\mathbf{Q}}$ as weakly stationary, then*

$$\sum_{i=1}^{Q} |\mu_i - \bar{\mu}_{1:Q}| = \left(\sqrt{40}c + 2\right)^{2/3} \log^{1/3}(T)Q^{2/3}\mathcal{V}_{1:Q}^{1/3}.$$

*(3) if it classifies the sequence $\mu^{\mathbf{Q}}$ as non-stationary at round $S$, then*

$$\sum_{i=1}^{S-1} |\mu_i - \bar{\mu}_{1:S-1}| = \left(\sqrt{40}c + 2\right)^{2/3} \log^{1/3}(T)S^{2/3}\mathcal{V}_{1:S}^{1/3}.$$

*Proof.* To prove (1), we assume that $\mu^{\mathbf{Q}}$ is weakly stationary (that is, $\mathcal{V}_{1:n} \leq 1/\sqrt{n}$ for any $n \in [Q]$), and upper-bound $\left|\bar{X}_{n_1:n_2} - \bar{X}^c_{n_1:n_2}\right|$. Thus, set some $n \in [Q]$ and $n_1, n_2 \in [n]$.

$$
\begin{aligned}
\left|\bar{X}_{n_1:n_2} - \bar{X}^c_{n_1:n_2}\right| &= \left|\bar{X}_{n_1:n_2} - \bar{X}^c_{n_1:n_2} - \bar{\mu}_{n_1:n_2} + \bar{\mu}^c_{n_1:n_2} + \bar{\mu}_{n_1:n_2} - \bar{\mu}^c_{n_1:n_2}\right| \\
&\leq \left|\bar{X}_{n_1:n_2} - \bar{X}^c_{n_1:n_2} - \bar{\mu}_{n_1:n_2} + \bar{\mu}^c_{n_1:n_2}\right| + \left|\bar{\mu}_{n_1:n_2} - \bar{\mu}^c_{n_1:n_2}\right| \\
&\overset{(a)}{\leq} \sqrt{2.5}c \log^{1/2}(T)\varepsilon_2(n_1, n_2) + \log^{1/2}(T)\varepsilon_2(n_1, n_2) \\
&= \left(\sqrt{2.5}c + 1\right)\log^{1/2}(T)\varepsilon_2(n_1, n_2),
\end{aligned}
$$

where (a) holds since $\mathbf{X}^{\mathbf{Q}}$ is strongly concentrated and $\mu^{\mathbf{Q}}$ is weakly stationary. It follows that TEST 2 classifies $\mu^{\mathbf{n}}$ as weakly stationary.

To prove (2), assume that TEST 2 classified $\mu^{\mathbf{Q}}$ as weakly stationary. Thus, for any $n_1, n_2 \in [Q]$ it holds that

$$
\left|\bar{X}_{n_1:n_2} - \bar{X}^c_{n_1:n_2}\right| \leq \left(\sqrt{2.5}c + 1\right)\log^{1/2}(T)\varepsilon_2(n_1, n_2).
$$

Let $n_1, \ldots, n_m$ be rounds for which $\{\mu_{n_i} \geq \bar{\mu}_{1:Q}$ and $\mu_{n_i+1} < \bar{\mu}_{1:Q}\}$ or $\{\mu_{n_i} < \bar{\mu}_{1:Q}$ and $\mu_{n_i+1} \geq \bar{\mu}_{1:Q}\}$. Additionally, denote $n_0 = 0$ and $n_{m+1} = n$. Now, the term we are interested in bounding can be written equivalently as follows

$$
\begin{aligned}
\sum_{i=1}^{Q} \left|\mu_i - \bar{\mu}_{1:Q}\right| &= \sum_{i=0}^{m} (n_{i+1} - n_i)\left|\bar{\mu}_{n_i+1:n_{i+1}} - \bar{\mu}_{1:Q}\right| \\
&= \sum_{i=0}^{m} (n_{i+1} - n_i)\left|\bar{\mu}_{n_i+1:n_{i+1}} - \bar{\mu}_{1:Q}\right|^{1/3}\left|\bar{\mu}_{n_i+1:n_{i+1}} - \bar{\mu}_{1:Q}\right|^{2/3}. \quad (7)
\end{aligned}
$$

Next, we bound the terms of the form $\left|\bar{\mu}_{n_i+1:n_{i+1}} - \bar{\mu}_{1:Q}\right|$. By the triangle inequality it holds that

$$
\left|\bar{\mu}_{n_i+1:n_{i+1}} - \bar{\mu}_{1:Q}\right| \leq \left|\bar{\mu}_{n_i+1:n_{i+1}} - \bar{X}_{n_i+1:n_{i+1}}\right| + \left|\bar{X}_{n_i+1:n_{i+1}} - \bar{X}_{1:Q}\right| + \left|\bar{X}_{1:Q} - \bar{\mu}_{1:Q}\right|.
$$

Since $\mathbf{X}^{\mathbf{Q}}$ is concentrated, we know that $\left|\bar{\mu}_{n_i+1:n_{i+1}} - \bar{X}_{n_i+1:n_{i+1}}\right| \leq \sqrt{2.5}c \log^{1/2}(T)\varepsilon_1(n_i + 1, n_{i+1})$, and also that $\left|\bar{\mu}_{1:Q} - \bar{X}_{1:Q}\right| \leq \sqrt{2.5}c \log^{1/2}(T)\varepsilon_1(1, Q)$. Trivially, it holds that $\varepsilon_1(1, Q) \leq \varepsilon_1(n_i + 1, n_{i+1})$. The third term requires a somewhat longer argument. Notice we can write

$$
\begin{aligned}
\left|\bar{X}_{n_i+1:n_{i+1}} - \bar{X}_{1:Q}\right| &= \left|\frac{1}{n_{i+1} - n_i}\sum_{j=n_i+1}^{n_{i+1}} X_j - \frac{1}{Q}\sum_{j \in [Q]} X_j\right| \\
&= \left|\frac{1}{n_{i+1} - n_i}\sum_{j=n_i+1}^{n_{i+1}} \left(X_j - \frac{n_{i+1} - n_i}{Q}X_j\right) - \frac{1}{Q - n_{i+1} + n_i}\sum_{j \notin \{n_i+1,\ldots,n_{i+1}\}} \frac{Q - n_{i+1} + n_i}{Q}X_j\right| \\
&= \left|\frac{1}{n_{i+1} - n_i}\sum_{j=n_i+1}^{n_{i+1}} \frac{Q - n_{i+1} + n_i}{Q}X_j - \frac{1}{Q - n_{i+1} + n_i}\sum_{j \notin \{n_i+1,\ldots,n_{i+1}\}} \frac{Q - n_{i+1} + n_i}{Q}X_j\right| \\
&= \left|\bar{X}_{n_i+1:n_{i+1}} - \bar{X}^c_{n_i+1:n_{i+1}}\right|\frac{Q - n_{i+1} + n_i}{Q}.
\end{aligned}
$$

Now, since the test classified the sequence as weakly stationary we know that $\left|\bar{X}_{n_i+1:n_{i+1}} - \bar{X}^c_{n_i+1:n_{i+1}}\right| \leq \left(\sqrt{2.5}c + 1\right)\log^{1/2}(T)\varepsilon_2(n_1, n_2)$, which implies that

$$
\left|\bar{X}_{n_i+1:n_{i+1}} - \bar{X}_{1:Q}\right| \leq \left(\sqrt{2.5}c + 1\right)\log^{1/2}(T)\left(\frac{Q - n_{i+1} + n_i}{Q}\right)\varepsilon_2(n_i + 1, n_{i+1}).
$$

By substituting the value of $\varepsilon_2(n_i + 1, n_{i+1})$ we get that

$$\left|\bar{X}_{n_i+1:n_{i+1}} - \bar{X}_{1:Q}\right| \leq \left(\sqrt{2.5}c + 1\right) \log^{1/2}(T) \left(\frac{Q - n_{i+1} + n_i}{Q}\right) \left(\frac{1}{n_{i+1} - n_i} + \frac{1}{Q - n_{i+1} + n_i}\right)^{1/2}$$

$$\leq \left(\sqrt{2.5}c + 1\right) \log^{1/2}(T) \left(\frac{Q - n_{i+1} + n_i}{Q}\right) \left(\frac{1}{n_{i+1} - n_i}\right)^{1/2}$$

$$+ \left(\sqrt{2.5}c + 1\right) \log^{1/2}(T) \left(\frac{Q - n_{i+1} + n_i}{Q}\right) \left(\frac{1}{Q - n_{i+1} + n_i}\right)^{1/2}$$

$$\leq \left(\sqrt{2.5}c + 1\right) \log^{1/2}(T) \left(\frac{1}{n_{i+1} - n_i}\right)^{1/2}$$

$$+ \left(\sqrt{2.5}c + 1\right) \log^{1/2}(T) \left(\frac{Q - n_{i+1} + n_i}{Q^2}\right)^{1/2}$$

$$\leq \left(\sqrt{2.5}c + 1\right) \log^{1/2}(T) \left(\frac{1}{n_{i+1} - n_i}\right)^{1/2} + \left(\sqrt{2.5}c + 1\right) \log^{1/2}(T) \left(\frac{1}{Q}\right)^{1/2}$$

$$\leq \left(\sqrt{10}c + 1\right) \log^{1/2}(T) \left(\frac{1}{n_{i+1} - n_i}\right)^{1/2}$$

$$\leq \left(\sqrt{10}c + 1\right) \log^{1/2}(T)\varepsilon_1(n_i + 1, n_{i+1}).$$

Combining all three calculations gives $\left|\bar{\mu}_{n_i+1:n_{i+1}} - \bar{\mu}_{1:Q}\right| \leq \left(\sqrt{40}c+2\right) \log^{1/2}(T)\varepsilon_1(n_i+1, n_{i+1})$. Substituting this in Eq. (7) yields

$$\sum_{i=1}^{Q} |\mu_i - \bar{\mu}_{1:Q}| \leq \sum_{i=0}^{m} (n_{i+1} - n_i) \left|\bar{\mu}_{n_i+1:n_{i+1}} - \bar{\mu}_{1:Q}\right|^{1/3} \left(\sqrt{40}c + 2\right)^{2/3} \log^{1/3}(T)\varepsilon_1(n_i + 1, n_{i+1})^{2/3}$$

$$= \left(\sqrt{40}c + 2\right)^{2/3} \log^{1/3}(T) \sum_{i=0}^{m} (n_{i+1} - n_i) \left|\bar{\mu}_{n_i+1:n_{i+1}} - \mu_{1:Q}\right|^{1/3} (n_{i+1} - n_i)^{-1/3}$$

$$\leq \left(\sqrt{40}c + 2\right)^{2/3} \log^{1/3}(T) \sum_{i=0}^{m} (n_{i+1} - n_i)^{2/3} \left|\bar{\mu}_{n_i+1:n_{i+1}} - \mu_{1:Q}\right|^{1/3}$$

$$\stackrel{(a)}{\leq} \left(\sqrt{40}c + 2\right)^{2/3} \log^{1/3}(T) \left(\sum_{i=0}^{m} (n_{i+1} - n_i)\right)^{2/3} \left(\sum_{i=0}^{m} \left|\bar{\mu}_{n_i+1:n_{i+1}} - \mu_{1:Q}\right|\right)^{1/3}$$

$$\stackrel{(b)}{\leq} \left(\sqrt{40}c + 2\right)^{2/3} \log^{1/3}(T) \left(\sum_{i=0}^{m} (n_{i+1} - n_i)\right)^{2/3} \left(\sum_{i=1}^{m} \left|\bar{\mu}_{n_i+1:n_{i+1}} - \bar{\mu}_{n_{i-1}+1:n_i}\right|\right)^{1/3}$$

$$\stackrel{(c)}{\leq} \left(\sqrt{40}c + 2\right)^{2/3} \log^{1/3}(T) Q^{2/3} \mathcal{V}_{1:Q}^{1/3}$$

where (a) follows from Hölder's inequality; (b) holds since $\left(\bar{\mu}_{n_i+1:n_{i+1}} - \mu_{1:Q}\right)$ and $\left(\bar{\mu}_{n_{i-1}+1:n_i} - \mu_{1:Q}\right)$ have different signs, according to the definition of $n_i$. Inequality (c) follows from Claim B.3 and the fact that $\sum_{i=1}^{m} \mathcal{V}_{n_{i-1}+1:n_{i+1}} \leq \mathcal{V}_{1:n}$.

The proof of (3) resembles the proof of (2) in its techniques, but is somewhat more subtle. Thus, denote by $S$ the round in which TEST 2 classified the sequence as non-stationary, and let $n_1, \ldots, n_m$ be rounds for which $\{\mu_{n_i} \geq \bar{\mu}_{1:S-1}$ and $\mu_{n_i+1} < \bar{\mu}_{1:S-1}\}$ or $\{\mu_{n_i} < \bar{\mu}_{1:S-1}$ and $\mu_{n_i+1} \geq \bar{\mu}_{1:S-1}\}$. Additionally, denote $n_0 = 0$ and $n_{m+1} = S$. Now, the term we are interested in can be written as follows:

$$\sum_{i=1}^{S-1} |\mu_i - \bar{\mu}_{1:S-1}| = \sum_{i=0}^{m} (n_{i+1} - n_i) \left|\bar{\mu}_{n_i+1:n_{i+1}} - \bar{\mu}_{1:S-1}\right|$$

$$= \sum_{i=0}^{m} (n_{i+1} - n_i) \left|\bar{\mu}_{n_i+1:n_{i+1}} - \bar{\mu}_{1:S-1}\right|^{1/3} \left|\bar{\mu}_{n_i+1:n_{i+1}} - \bar{\mu}_{1:S-1}\right|^{2/3}.$$

As for the proof of part (2), it holds that $\left|\bar{\mu}_{n_i+1:n_{i+1}} - \mu_{1:S-1}\right| \leq \left(\sqrt{40}c + 2\right)\log^{1/2}(T)\varepsilon_1(n_i + 1, n_{i+1})$. Substituting this above gives

$$\sum_{i=1}^{S-1}|\mu_i - \bar{\mu}_{1:S-1}| \leq \sum_{i=0}^{m}(n_{i+1} - n_i)\left|\bar{\mu}_{n_i+1:n_{i+1}} - \bar{\mu}_{1:S-1}\right|^{1/3}\left(\sqrt{40}c + 2\right)^{2/3}\log^{1/3}(T)\varepsilon_1(n_i + 1, n_{i+1})^{2/3}$$

$$\leq \left(\sqrt{40}c + 2\right)^{2/3}\log^{1/3}(T)S^{2/3}\mathcal{V}_{1:S}^{1/3},$$

where the same arguments from the proof of (2) apply here also. $\qquad\square$

## C Analysis of the Stochastic Feedback Setting

In this section we provide an analysis of the performance of our main algorithm. In our analysis we partition the randomness into two independent sources. The first is that originating from the noisy observations $\ell_t(i)$, combined with the source of random bits based on which the algorithm determines the realizations of the random variables $X_t$. The second source of randomness is that determining during the exploitation phase whether we pick $\arg\min\{\hat{\mu}_0(i)\}$ or not, in each bin. Notice that the first source can be thought of as if we generate a random sequence $\gamma^{\mathbf{T}} \in \{1,2\}^T$ beforehand, and define two new sequences:

$$X_t(i) = \begin{cases} 2\ell_t(\gamma_t) & \text{if } i = \gamma_t \\ 0 & \text{otherwise.} \end{cases}$$

Thus, we overall have four sequences ($\{\ell_t(i)\}_{t=1}^T$ and $\{X_t(i)\}_{t=1}^T$, for $i = 1, 2$), that encapsulate the first source of randomness. During the analysis we condition on this source, and assume it is such that all four sequences are strongly concentrated. By Claim 2.3 we know that this event occurs w.p. of at least $1 - \frac{4}{T}$.

The second source of randomness can also be controlled beforehand, exactly as detailed in the proof of Theorem A.1. Essentially, we show that with high probability the worst arm is picked throughout at most $\log(T)N_j^{1-\theta}$ bins in block $B_j$. In addition, we show that the number of "bad" bins we encounter in block $B_j$ is at most $\log(T)N_j^{\theta}$. The definition of these "bad" bins (non-flat bins), along with the proofs of these two observation are given below.

**Definition C.1.** *(flat bin) We say that a bin $A$ is* flat *if either*

$$\sum_{t \in A}|\mu_t(i) - \hat{\mu}_0(i)| \leq \left(\sqrt{10} + 2\right)^{2/3}\log^{1/3}(T)|A|^{2/3}\mathcal{V}_A^{1/3}$$

*or*

$$|\mu_t(i) - \hat{\mu}_0(i)| \leq 2\left(\sqrt{10} + 2\right)\log^{1/2}(T)|A|^{-1/2}, \ \forall t \in A$$

*for any action $i \in \{1,2\}$; otherwise we say it is* non-flat.

From hereon, whenever $j$ is set and block $B_j$ is considered, we denote by $\delta_a$ the indicator of the event {the $a$-th bin in block $B_j$ is non-flat}.

**Lemma C.2.** *Assuming $\theta \leq 1/2$, the following event occurs with probability at least $1 - \frac{1}{T}$. For any block $B_j$ that reached the exploitation phase, with $N_j$ bins in total, the total number of bins in which the worst arm (according to the estimates of the exploration phase) is chosen is upper bounded by $\frac{5\log^{1/2}(T)}{(1-\theta)}N_j^{1-\theta}$.*

*Proof.* Fix some block $B_j$. Notice that given the realization of the source of randomness determining our actions in the exploration phase and the observed losses, both $\Delta, \hat{\mu}_0(i)$ for $i = \{1,2\}$, and the starting point for the exploitation phase are determined only based on the starting time $t_{\text{start}}$ of the block. As a result, the lengths of the bins and the flatness of each bin is also deterministic given the mentioned source of randomness and $t_{\text{start}}$. We prove a high probability result for any starting time and value $N_j$ and obtain the result via a union bound argument.

Define $I_a$ as the indicator of the event {the worst arm was picked throughout bin $A_{j,a}$}. The expected value of $I_t$ is $\frac{1}{2}a^{-\theta}$, hence

$$\mathbb{E}\left[\sum_{t=1}^{n} I_t\right] = \sum_{t=1}^{n}\frac{1}{2}a^{-\theta} \le \frac{1}{2(1-\theta)}n^{1-\theta},$$

for any $n \in [T]$. Setting $\varepsilon = \frac{4\log^{1/2}(T)}{1-\theta}n^{-\theta}$ gives

$$\Pr\left(\sum_{t=1}^{n} X_t > \frac{1}{2(1-\theta)}n^{1-\theta} + \frac{4\log^{1/2}(T)}{1-\theta}n^{1-\theta}\right) \le e^{-2n\varepsilon^2} \le 1/T^4,$$

where the last inequality holds since $2n\varepsilon^2 = \frac{16\log(T)}{(1-\theta)^2}n^{1-2\theta} \ge 4\log(T)$ for $\theta \le 1/2$. As there are at most $T$ possible values of both $n$ and $t_{\text{start}}$, the result follows via union bound. $\qquad\square$

**Lemma C.3.** *The following event occurs w.p. at least $1 - \frac{1}{T}$. For any $j$, in which Algorithm 1 reached the exploitation phase in block $B_j$, the total number of non-flat bins in this block is upper-bounded by $9\log(T)N_j^\theta$.*

*Proof.* Set $j$ and look at block $B_j$. Let $\delta_a$ be as defined before, and denote by $\{a_n\}_{n=1}^\infty$ the bins in which $\delta_{a_n} = 1$. Notice that given the realization of the source of randomness determining our actions in the exploration phase and the observed losses, both $\Delta$, $\hat{\mu}_0(i)$ for $i = \{1,2\}$, and the starting point for the exploitation phase are determined only based on the starting time $t_{\text{start}}$ of the block. As a result, the sequence $\{a_n\}_{n=1}^\infty$ is also deterministic given the mentioned source of randomenss and $t_{\text{start}}$.

Define

$$s_{\min} = \arg\min_s\left\{\sum_{n=1}^{s} a_n^{-\theta} \ge 8\log(T)\right\}.$$

We have that for a fixed $t_{\text{start}}$,

$$\log\left(\mathbb{P}\left(|N_j| > a_{s_{\min}}\right)\right) \le \log\prod_{n=1}^{s_{\min}}\left(1 - \frac{1}{2}a_n^{-\theta}\right) = \sum_{n=1}^{s_{\min}}\log\left(1 - \frac{1}{2}a_n^{-\theta}\right) \le \sum_{n=1}^{s_{\min}}\left(-\frac{1}{2}a_n^{-\theta}\right) \le -4\log(T),$$

where we use the fact that when $\delta_a = 1$, the probability of stopping is at least $\frac{1}{2}a_n^{-\theta}$ by Algorithm 1. It follows that $\mathbb{P}\left(|N_j| \le a_{s_{\min}}\right) \ge 1 - \frac{1}{T^4}$. Now, let $Y_j$ denote the total number of bins in block $B_j$ for which $\delta_a = 1$. From the calculation above we know that

$$(Y_j - 1)a_{Y_j}^{-\theta} \le \sum_{n=1}^{Y_j-1} a_n^{-\theta} < 8\log(T),$$

or equivalently, $Y_j < 1 + 8\log(T)a_{Y_j}^\theta \le 9\log(T)N_j^\theta$. If $s_{\min}$ does not exist, then it must hold that $\sum_{n=1}^{s} a_n^{-\theta} < 8\log(T)$ for any $s$ and in particular for $Y_j$. Using the argument above gives $Y_j \le 8\log(T)N_j^\theta$ for this case.

By taking a union bound over all possible $t_{\text{start}}$, it follows that with probability of at least $1 - \frac{1}{T}$ we have that the number of non-flat bins in any block $B_j$ is upper-bounded by $9\log(T)N_j^\theta$. $\qquad\square$

After establishing these two lemmas, we condition the game on the high probability events described in them. We use this at the end of the section to provide a high probability bound on the overall regret. We begin by bounding the regret in the blocks where we never reach the exploitation phase. We proceed to refine the bound via a lower bound on the variation in these blocks. To ease the notation, we denote $\bar{\mu}_D(i) = \frac{1}{|D|}\sum_{t\in D}\mu_t(i)$ and $\bar{X}_D(i) = \frac{1}{|D|}\sum_{t\in D}X_t(i)$, where $D$ can be any subset of rounds (in particular, a block or a bin). In addition, we denote by $\mathcal{R}_D$ the regret suffered in rounds $t \in D$. That is, $\mathcal{R}_D = \sum_{t\in D}\mu_t(i_t) - \mu_t(i_t^*)$. Thus, fix a block index $j$ and recall that $E_{j,1}$ denotes the set of round indices in the exploration phase of the block and $\mathcal{V}_{E_{j,1}}$ denotes the variation in it.

**Lemma C.4.** *Fix a block index $j$ and consider its exploration phase. It holds that*

$$\mathcal{R}_{E_{j,1}} \leq 12 \log^{1/3}(T)|E_{j,1}|^{2/3}\mathcal{V}_{E_{j,1}}^{1/3} + 460 \log^{1/2}(T)|E_{j,1}|^{1-\lambda/2}.$$

*Proof.* We divide the analysis into two cases: (1) the exploration phase ended since the condition lower bounding $\Delta$ was met; and (2) the exploration phase ended since the test identified non-stationarity. If we are in the first case then from Claim B.4 part (2), by substituting $c$ by 2, we know that

$$\sum_{t \in E_{j,1}} |\mu_t(i) - \bar{\mu}_{E_{j,1}}(i)| \leq 6 \log^{1/3}(T)|E_{j,1}|^{2/3}\mathcal{V}_{E_{j,1}}^{1/3}$$

for both $i = 1, 2$. Thus,

$$\begin{aligned}
\sum_{t \in E_{j,1}} \mu_t(i_t) - \mu_t(i_t^*) &\leq \sum_{t \in E_{j,1}} |\mu_t(1) - \mu_t(2)| \\
&\stackrel{(a)}{\leq} \sum_{t \in E_{j,1}} |\mu_t(1) - \bar{\mu}_{E_{j,1}}(1)| + \sum_{t \in E_{j,1}} |\bar{\mu}_{E_{j,1}}(1) - \bar{X}_{E_{j,1}}(1)| \\
&\quad + \sum_{t \in E_{j,1}} |\bar{X}_{E_{j,1}}(1) - \bar{X}_{E_{j,1}}(2)| + \sum_{t \in E_{j,1}} |\bar{X}_{E_{j,1}}(2) - \bar{\mu}_{E_{j,1}}(2)| \\
&\quad + \sum_{t \in E_{j,1}} |\bar{\mu}_{E_{j,1}}(2) - \mu_t(2)| \\
&\stackrel{(b)}{\leq} \sum_{t \in E_{j,1}} |\bar{X}_{E_{j,1}}(1) - \bar{X}_{E_{j,1}}(2)| + 12 \log^{1/3}(T)|E_{j,1}|^{2/3}\mathcal{V}_{E_{j,1}}^{1/3} \\
&\quad + 7|E_{j,1}| \log^{1/2}(T)\varepsilon_1(1, |E_{j,1}|) \qquad (8)
\end{aligned}$$

where (a) follows by the triangle inequality; and (b) holds because the feedback is concentrated. Now, since Algorithm 1 did not stop at round $t(j, |E_{j,1}| - 1)$ we know that

$$\begin{aligned}
|\bar{X}_{E_{j,1}}(1) - \bar{X}_{E_{j,1}}(2)| &= |\bar{X}_{t(j,1):t(j,|E_{j,1}|)}(1) - \bar{X}_{t(j,1):t(j,|E_{j,1}|)}(2)| \\
&\leq \frac{1}{|E_{j,1}|} + |\bar{X}_{t(j,1):t(j,|E_{j,1}|-1)}(1) - \bar{X}_{t(j,1):t(j,|E_{j,1}|-1)}(2)| \\
&\leq \frac{1}{|E_{j,1}|} + 16(\sqrt{10} + 2)^2 \log(T)(|E_{j,1}| - 1)^{-\lambda/2}.
\end{aligned}$$

Substituting the above in Eq. (8) yields the result stated in the lemma for the first case. The proof of the second case follows similarly to the first case, albeit using part (3) of Claim B.4 instead of part (2) and using the fact that $|\bar{X}_{E_{j,1}}(1) - \bar{X}_{E_{j,1}}(2)| \leq 16(\sqrt{10} + 2)^2 \log(T)|E_{j,1}|^{-\lambda/2}$ for this case. $\square$

**Lemma C.5.** *The total regret suffered throughout blocks in which Algorithm 1 did not reach the exploitation phase is upper-bounded by*

$$12 \log^{1/3}(T)T^{2/3}\mathcal{V}_T^{1/3} + 460 \log(T)T^{1-\lambda/3}\mathcal{V}_T^{\lambda/3} + 460 \log(T)T^{1-\lambda/2}.$$

*Proof.* By Claim B.4 part (1) we know that the algorithm never misclassifies a sequence as non-stationary during the exploration phase (since the the feedback to the test is strongly concentrated). Now, let

$$J_2 = \{j \mid \text{TEST } 2 \text{ identified non-stationarity during the exploration of block } B_j\}.$$

Using this notation, we know that for every $j \in J_2$ it holds that $\mathcal{V}_{B_j} \geq |B_j|^{-1/2}$. In addition, by Lemma C.4 we know that for every $j \in J_2$ it holds that

$$\mathcal{R}_{B_j} \leq 12 \log^{1/3}(T)|B_j|^{2/3}\mathcal{V}_{B_j}^{1/3} + 460 \log(T)|B_j|^{1-\lambda/2}.$$

Thus, we can write

$$\sum_{j\in J_2} \mathcal{R}_{B_j} \leq 12\log^{1/3}(T)\sum_{j\in J_2}|B_j|^{2/3}\mathcal{V}_{B_j}^{1/3} + 460\log(T)\sum_{j\in J_2}|B_j|^{1-\lambda/2}$$

$$= 12\log^{1/3}(T)\sum_{j\in J_2}|B_j|^{2/3}\mathcal{V}_{B_j}^{1/3} + 460\log(T)\sum_{j\in J_2}|B_j|^{1-\lambda/3}|B_j|^{-\lambda/6}$$

$$\overset{(a)}{\leq} 12\log^{1/3}(T)\left(\sum_{j\in J_2}|B_j|\right)^{2/3}\left(\sum_{j\in J_2}\mathcal{V}_{B_j}\right)^{1/3}$$

$$+ 460\log(T)\left(\sum_{j\in J_2}|B_j|\right)^{1-\lambda/3}\left(\sum_{j\in J_2}|B_j|^{-1/2}\right)^{\lambda/3}$$

$$\leq 12\log^{1/3}(T)T^{2/3}\mathcal{V}_T^{1/3} + 460\log(T)T^{1-\lambda/3}\mathcal{V}_T^{\lambda/3},$$

where (a) follows by Hölder's inequality. Notice that $J_2$ might not include the last block in the game (even if Algorithm 1 did not reach the exploitation phase in this block), simply because this block can be terminated due to the end of the time horizon. Thus, an extra term $460\log(T)|B_j|^{1-\lambda/2} \leq 460\log(T)T^{1-\lambda/2}$ is added to the regret in this case. $\qquad\square$

We now proceed to bound the regret in the blocks in which the exploitation phase is reached. We begin by providing a bound with an additive element dependent only on the block size and not its variation, and will prove later on a satisfactory bound on this quantity.

**Lemma C.6.** *Set $j$, and assume that Algorithm 1 reached the exploitation phase in block $B_j$. Then, for any flat bin $A_{j,a} \in B_j$ in which the best arm was chosen it holds that*

$$\sum_{t\in A_{j,a}} \mu_t(i_t) - \mu_t(i_t^*) \leq 6\log^{1/3}(T)|A_{j,a}|^{2/3}\mathcal{V}_{A_{j,a}}^{1/3}.$$

*Proof.* Assume without loss of generality that action 1 was the better action in the exploration phase, that is, $1 = \arg\min_{i\in\{1,2\}}\{\hat{\mu}_0(i)\}$. Thus, we have

$$\sum_{t\in A_{j,a}} \mu_t(i_t) - \mu_t(i_t^*) = \sum_{t\in A_{j,a}} (\mu_t(1) - \mu_t(i_t^*)) = \sum_{t\in A_{j,a}, i_t^*=2} (\mu_t(1) - \mu_t(2)).$$

In order to bound $\sum_{i_t^*=2}(\mu_t(1) - \mu_t(2))$ we use the guarantee of the bin *flatness*, and divide the analysis into three cases: In the first case we assume that

$$\sum_{t\in A_{j,a}}|\mu_t(i) - \hat{\mu}_0(i)| \leq \left(\sqrt{10}+2\right)^{2/3}\log^{1/3}(T)|A_{j,a}|^{2/3}\mathcal{V}_{A_{j,a}}^{1/3}$$

for $i \in \{1,2\}$. Thus, by Claim B.2 part (2) we have that

$$\sum_{t\in A_{j,a}, i_t^*=2}(\mu_t(1) - \mu_t(2)) = \sum_{t\in A_{j,a}, i_t^*=2}(\mu_t(1) - \hat{\mu}_0(1) + \hat{\mu}_0(1) - \hat{\mu}_0(2) + \hat{\mu}_0(2) - \mu_t(2))$$

$$\leq \sum_{t\in A_{j,a}}|\mu_t(1) - \hat{\mu}_0(1)| + \sum_{t\in A_{j,a}}|\mu_t(2) - \hat{\mu}_0(2)| + \sum_{t\in A_{j,a}, i_t^*=2}(\hat{\mu}_0(1) - \hat{\mu}_0(2))$$

$$\leq 2\left(\sqrt{10}+2\right)^{2/3}\log^{1/3}(T)|A_{j,a}|^{2/3}\mathcal{V}_{A_{j,a}}^{1/3} + \sum_{t\in A_{j,a}, i_t^*=2}(\hat{\mu}_0(1) - \hat{\mu}_0(2)).$$

Now, since action 1 is assumed to be the better action (in the exploration phase of the block), then it must hold that $\sum_{t\in A_{j,a}, i_t^*=2}(\hat{\mu}_0(1) - \hat{\mu}_0(2)) \leq 0$, which proves the lemma for this case.

In the second case we assume that $|\mu_t(i) - \hat{\mu}_0(i)| \leq 2\left(\sqrt{10}c + 2\right)\log^{1/2}(T)|A_{j,a}|^{-1/2}$ for any $t \in A_{j,a}$ and $i \in \{1,2\}$. Notice that the term $(\hat{\mu}_0(1) - \hat{\mu}_0(2))$ can be bounded as follows:

$$\hat{\mu}_0(1) - \hat{\mu}_0(2) = \bar{X}_{E_{j,1}}(1) - \bar{X}_{E_{j,1}}(2)$$

$$\leq -16\left(\sqrt{10}+2\right)^2\log(T)|E_{j,1}|^{-\lambda/2}$$

$$\leq -4\left(\sqrt{10}+2\right)\log^{1/2}(T)|A_{j,a}|^{-1/2},$$

hence it must be the case that for all $i \in \{1, 2\}$ and $t \in A_{j,a}$

$$\mu_t(1) \le \hat{\mu}_0(1) + 2(\sqrt{10} + 2) \log^{1/2}(T)|A_{j,a}|^{-1/2}$$
$$\le \hat{\mu}_0(2) - 2(\sqrt{10} + 2) \log^{1/2}(T)|A_{j,a}|^{-1/2} \le \mu_t(2).$$

Therefore, there are no rounds $t \in A_{j,a}$ in which $i_t^* = 2$, hence $\sum_{t \in A_{j,a}, i_t^* = 2} (\mu_t(1) - \mu_t(2)) = 0$.

In the third case we have w.l.o.g. that for $i = 1$

$$\sum_{t \in A_{j,a}} |\mu_t(1) - \hat{\mu}_0(1)| \le (\sqrt{10} + 2)^{2/3} \log^{1/3}(T)|A_{j,a}|^{2/3} \mathcal{V}_{A_{j,a}}^{1/3},$$

and for $i = 2$

$$|\mu_t(2) - \hat{\mu}_0(2)| \le 2(\sqrt{10} + 2) \log^{1/2}(T)|A_{j,a}|^{-1/2}, \ \forall t \in A_{j,a}.$$

Using the bound above for $\hat{\mu}_0(1) - \hat{\mu}_0(2)$ we get that for all $t \in A_{j,a}$ it holds that $\mu_t(2) \ge \hat{\mu}_0(1)$, hence in any round where $i_t^* = 2$ we have $\mu_t(1) - \mu_t(2) \le \mu_t(1) - \hat{\mu}_0(1)$ leading to

$$\sum_{t \in A_{j,a}, i_t^* = 2} (\mu_t(1) - \mu_t(2)) \le \sum_{t \in A_{j,a}, i_t^* = 2} (\mu_t(1) - \hat{\mu}_0(1))$$
$$\le \sum_{t \in A_{j,a}} |\mu_t(1) - \hat{\mu}_0(1)|$$
$$\le (\sqrt{10}c + 2)^{2/3} \log^{1/3}(T)|A_{j,a}|^{2/3} \mathcal{V}_{A_{j,a}}^{1/3}.$$

$\square$

To bound the regret throughout flat bins in which the worse arm was picked, we simply use the guarantee presented in Lemma C.2 on the total amount of such bins.

**Lemma C.7.** *Set $j$, and assume that Algorithm 1 reached the exploitation phase in block $B_j$. Then, the total regret suffered throughout flat bins in which the worst arm was chosen is at most*

$$\frac{5 \log^{1/2}(T)}{(1 - \theta)}|B_j|^{1 - \theta(1 - \lambda)}$$

*Proof.* From Lemma C.2 we know that the total number of flat bins in which the worst arm was chosen is upper-bounded by

$$\frac{5 \log^{1/2}(T)}{(1 - \theta)} N_j^{1 - \theta}.$$

In each of these bins, the regret we suffer is naturally at most $|A_{j,a}|$. Thus, we can bound the total regret as follows:

$$\frac{5 \log^{1/2}(T)}{(1 - \theta)} N_j^{1 - \theta}|A_{j,a}| = \frac{5 \log^{1/2}(T)}{(1 - \theta)}|A_{j,a}| \left( \frac{|E_{j,2}|}{|A_{j,a}|} \right)^{1 - \theta} = \frac{5 \log^{1/2}(T)}{(1 - \theta)}|E_{j,2}|^{1 - \theta}|A_{j,a}|^\theta.$$

Now, since $|E_{j,2}| \le |B_j|$ and $|A_{j,a}| \le |E_{j,1}|^\lambda \le |B_j|^\lambda$ by the algorithm (for blocks in which the exploitation phase was reached), we overall get the result stated in the lemma. $\square$

**Lemma C.8.** *Set $j$, and assume that Algorithm 1 reached the exploitation phase in block $B_j$. Then, the total regret suffered in flat bins throughout block $B_j$ is bounded by*

$$\sum_{a=1}^{N_j} (1 - \delta_a) \mathcal{R}_{A_{j,a}} \le 6 \log^{1/3}(T)|B_j|^{2/3} \mathcal{V}_{B_j}^{1/3} + \frac{5 \log^{1/2}(T)}{(1 - \theta)}|B_j|^{1 - \theta(1 - \lambda)}.$$

*Proof.* The result follows via simple aggregation of Lemmas C.6 and C.7, and simple application of Hölder's inequality. $\square$

We proceed to bound the regret suffered due to non-flat bins. Here, rather than bound the regret inside such bins we bound the total number of such bins.

**Lemma C.9.** *Set $j$, and assume that Algorithm 1 reached the exploitation phase in block $B_j$. Then, the total regret suffered in non-flat bins throughout block $B_j$ is bounded by*

$$\sum_{a=1}^{N_j} \delta_a \mathcal{R}_{A_{j,a}} \leq 9 \log(T)|B_j|^{1-(1-\theta)(1-\lambda)}.$$

*Proof.* Set $j$ and look at block $B_j$. Since we condition the game on the high probability event of Lemma C.3 we know that the number of non-flat bins in this block is upper-bounded by $9\log(T)N_j^\theta$. In each of these bins, the regret suffered is upper-bounded by the length of the bin, $|A_{j,a}|$. Thus, by summing over all non-flat bins we get that

$$\sum_{a=1}^{N_j} \delta_a \mathcal{R}_{A_{j,a}} \leq 9 \log(T)|A_{j,a}|N_j^\theta = 9 \log(T)|A_{j,a}| \left( \frac{|E_{j,2}|}{|A_{j,a}|} \right)^\theta = 9 \log(T)|E_{j,2}|^\theta |A_{j,a}|^{1-\theta}.$$

Now, since $|E_{j,2}| \leq |B_j|$ and $|A_{j,a}| \leq |E_{j,1}|^\lambda \leq |B_j|^\lambda$ by the algorithm (for blocks in which the exploitation phase was reached), we overall get the result stated in the lemma. $\qquad\square$

From the above we get a bound on the regret inside each block. The bound has an additive term dependent only on $|B_j|$ and not $\mathcal{V}_{B_j}$ that does not trivially add up to a sub-linear term in $T$. In the following we use a lower bound on $\mathcal{V}_{B_j}$ to deal with this issue, and get our final regret bound for blocks in which the exploitation phase is reached.

**Lemma C.10.** *The total regret suffered throughout blocks in which Algorithm 1 reached the exploitation phase is upper-bounded by*

$$6 \log^{1/3}(T)T^{2/3}\mathcal{V}_T^{1/3} + 20 \log^{1/2}(T)T^{\frac{1+2\lambda}{2+\lambda}}\mathcal{V}_T^{\frac{1-\lambda}{2+\lambda}} + 20\log(T)T^{1-\frac{1}{2}(1-\lambda)},$$

*if we set $\theta = \frac{1}{2}$.*

*Proof.* By Claim B.2 part (1) we know that the algorithm never misclassifies a sequence as non-stationary during the exploration phase (if the feedback is concentrated). Now, let

$$J_1 = \{j \mid \text{TEST 1 identified non-stationarity during the exploitation of block } B_j\}.$$

Denote by $|A_{j,a}|$ the bin length in block $B_j$. Using this notation, we know that for every $j \in J_1$ it holds that $\mathcal{V}_{B_j} \geq |A_{j,a}|^{-1/2} \geq |E_{j,1}|^{-\lambda/2} \geq |B_j|^{-\lambda/2}$. From Lemmas C.8 and C.9, we obtain that for every $j \in J_1$ we can upper-bound

$$\mathcal{R}_{B_j} \leq 6 \log^{1/3}(T)|B_j|^{2/3}\mathcal{V}_{B_j}^{1/3} + 20\log(T)|B_j|^{1-\frac{1}{2}(1-\lambda)},$$

if $\theta = \frac{1}{2}$. Thus, we can write

$$\sum_{j \in J_1} \mathcal{R}_{B_j} \leq 8 \log^{1/3}(T) \sum_{j \in J_1} |B_j|^{2/3}\mathcal{V}_{B_j}^{1/3} + 8\log(T) \sum_{j \in J_1} |B_j|^{1-\frac{1}{2}(1-\lambda)}$$

$$= 6 \log^{1/3}(T) \sum_{j \in J_1} |B_j|^{2/3}\mathcal{V}_{B_j}^{1/3} + 20\log(T) \sum_{j \in J_1} |B_j|^{\frac{1+2\lambda}{2+\lambda}}|B_j|^{-\frac{\lambda(1-\lambda)}{2(2+\lambda)}}$$

$$\overset{(a)}{\leq} 6 \log^{1/3}(T) \left( \sum_{j \in J_1} |B_j| \right)^{2/3} \left( \sum_{j \in J_1} \mathcal{V}_{B_j} \right)^{1/3} + 20\log(T) \left( \sum_{j \in J_1} |B_j| \right)^{\frac{1+2\lambda}{2+\lambda}} \left( \sum_{j \in J_1} |B_j|^{-\lambda/2} \right)^{\frac{1-\lambda}{2+\lambda}}$$

$$\leq 6 \log^{1/3}(T)T^{2/3}\mathcal{V}_T^{1/3} + 20\log(T)T^{\frac{1+2\lambda}{2+\lambda}}\mathcal{V}_T^{\frac{1-\lambda}{2+\lambda}},$$

where (a) follows by Hölder's inequality. Notice that $J_1$ might not include the last block in the game (even if Algorithm 1 reached the exploitation phase in this block), simply because this block can be terminated due to the end of the time horizon. Thus, an extra term $20 \log^{1/2}(T)|B_j|^{1-\frac{1}{2}(1-\lambda)} \leq 20\log^{1/2}(T)T^{1-\frac{1}{2}(1-\lambda)}$ needs to be added to the regret in this case. $\qquad\square$

We are now ready to combine the results above, and to prove Theorem 2.4.

*Proof of Theorem 2.4.* We start with the case where the feedback received throughout the game is strongly concentrated, and the high probability events of Lemmas C.2 and C.3 occur (as discussed in the beginning of the section). From Lemma C.5, we know that the total regret suffered throughout blocks in which Algorithm 1 did not reach the exploitation phase is upper-bounded by

$$12 \log^{1/3}(T) T^{2/3} \mathcal{V}_T^{1/3} + 460 \log(T) T^{1-\lambda/3} \mathcal{V}_T^{\lambda/3} + 460 \log(T) T^{1-\lambda/2}.$$

Additionally, from Lemma C.10 we know that the total expected regret suffered over blocks in which Algorithm 1 reached the exploitation phase is upper-bounded by

$$6 \log^{1/3}(T) T^{2/3} \mathcal{V}_T^{1/3} + 20 \log^{1/2}(T) T^{\frac{1+2\lambda}{2+\lambda}} \mathcal{V}_T^{\frac{1-\lambda}{2+\lambda}} + 20 \log(T) T^{1-\frac{1}{2}(1-\lambda)},$$

for the case where $\theta = \frac{1}{2}$. Summing the above together and substituting $\lambda = \frac{\sqrt{37}-5}{2}$ gives

$$\mathcal{R}_T = \sum_{t=1}^{T} \mu_t(i_t) - \sum_{t=1}^{T} \mu_t(i_t^*) \le 500 \log(T) T^{0.82} \mathcal{V}_T^{0.18} + 500 \log(T) T^{0.771}.$$

for this case. Now, recall that the probability of the feedback being strongly concentrated is $1 - \frac{4}{T}$ (Claim 2.3). Additionally, by taking union bound on the high probability events described in Lemmas C.2 and C.3, we get the upper bound above on the total regret (suffered throughout the entire game) with probability of at least $1 - \frac{10}{T}$.  □

# D  Extending the Result to $k$ Arms

Our results can be extended to the case of $k > 2$, using the same core techniques of the $k = 2$ case. Due to the technical nature of the algorithm and its analysis, we omit it from this version of the paper and defer it to a full version. The high-level idea is to replace the exploration phase of Algorithm 1 with a soft elimination tournament: Initially, the algorithm explores all arms (i.e., chooses each arm with probability of $1/k$ in each round). Once an arm is exhibiting significantly worse performance compared to the leading arm, this arm enters the pool of eliminated arms. From this point on, the algorithm works in bins with length proportional to the inverted squared gap between *the leading arm and the arm that was lastly eliminated*. The bin counter (denoted by $a$ in Algorithm 1) resets every time an arm is eliminated. Within each bin, the algorithm either:

(1) explores non-eliminated arms (i.e., chooses an arm from the set of non-eliminated arms uniformly in each round) with probability $1 - o(1)$.

(2) or otherwise samples one of the eliminated arms (and pulls it throughout the bin).

# E  Discussion and Conclusion

In this work we showed that the regret with respect to the optimal sequence of actions in the MAB setting can be non-trivially bounded if the environment is guaranteed to vary sufficiently slow. The important contribution over previous works that considered the same setting is the ability to adapt to the changes of the environment without requiring any prior knowledge on them whatsoever. An interesting open question is whether our bounds can be improved to match the current known lower bound for the problem $\Omega\big(T^{2/3} \mathcal{V}_T^{1/3}\big)$ [8]. It would also be interesting to extend our techniques to the BCO setting, in which the decision set is continuous and the losses are general convex functions.

## Footnotes

[5]If $s_{\min} = T$, then it must hold that $\sum_{n=1}^{s} \frac{1}{\sqrt{k\tau_n}} < 2\log(T)$ for any $s$ and in particular for $Y$.