[Reviews · NeurIPS 2016]

Reviewer 1

Summary

The paper considers a version of the nonstochastic bandit problem in which the player competes with the best action in each step, and the regret is expressed in terms of the sum V_T over time of the infinity norm of the difference between two consecutive loss vectors. In the noisy version of this problem, which is the main focus of the work, the player does not directly observe the loss of the chosen action, but rather the realization of a [0,1]-valued random variable whose expectation is equal to the loss. A previous paper showed that the optimal regret rate is T^{2/3}V_T^{1/3}, and analyzed an algorithm achieving this rate when V_T is known. In this work the authors introduce a different algorithm achieving a weaker (and possibly suboptimal) bound when V_T is unknown. The solution (for the noisy case) uses concentration to spot blocks of consecutive steps in which the losses vary too much to allow regret control. The algorithm explicitely distinguishes exploration and exploitation phases, and a different statistical test is used in each phase in order to detect nonstationarities.

Qualitative Assessment

The problem is interesting, even in the noiseless case. Neither the algorithm not its analysis (which does not seem to contain any striking new ideas) are well explained in the paper. Could the algorithm be run with lambda=0? This choice seems to give a good and interpretable regret bound. There might be connections with works that deal with the "best of both worlds" problem in the bandit setting (see, e.g., Bubeck and Slivkins in COLT 2012). Namely, algorithms that play optimally both in the stochastic and nonstochastic setting. They use similar statistical tests to assess the stationary/nonstationary nature of the loss sequence. The rate achieved in the noiseless case (which is achieved without knowledge of V_T) is essentially equivalent to the optimal rate for the noisy case. It would be interesting to understand what is the optimal rate for the noiseless case. In summary, the paper has the merit of providing a solution to an interesting and intriguing problem. On the other hand, few insights are provided about the techniques, and the algorithm is not particularly appealing. The choice of the parameters in the bound could perhaps be improved.

Confidence in this Review

2-Confident (read it all; understood it all reasonably well)


Reviewer 2

Summary

The paper analyzes a MAB problem with two parms in an adversarial setting and analyzes the design of algorithm that achieve sublinear regret with respect to the optimal sequence of actions (as opposed to the best single arm in hindsight). When an upper on the variation is known to be V_T and V_T= o(T), earlier work has shown that the minimax regret is given by O(T^{2/3}V_T^{1/3}). However, the algorithm previously proposed has access to V_T and can be tuned accordingly. In the present paper, the authors propose an algorithm that does not require knowledge of V_T and has regret of order O(T^{0.771} + T^{0.82} V_T^{0.18}).

Qualitative Assessment

I found the paper quite interesting and the analysis novel. On the improvements side, I found the paper would have been stronger with a lower bound that shows the potential price one needs to pay due to the lack of knowledge of the variation. At least an informed discussion of conjectures would be useful. I found the paper would also benefit from clarifying the formulation and in particular the space of actions of the adversary. In a minimax regret setting, the adversary will take V_T to be as high as possible. Is it assumed that the adversary has a constraint on the variation V_T (but is free except from that) and the algorithm designer does not know this constraint? A mathematical formulation of the full minimax problem (with proper action spaces and informational structure) would be helpful.

Confidence in this Review

2-Confident (read it all; understood it all reasonably well)


Reviewer 3

Summary

This paper considers multi-armed bandits with non-stationary stochastic rewards. The problem was considered in the reference [1] of this paper, but the solution was not adaptive, and hence, the adaptive solution was proposed as an open problem. This work develops an adaptive algorithm to tackle the problem and provides theoretical guarantees.

Qualitative Assessment

I believe that the adaptive solution to multi-armed bandits with non-stationary rewards is challenging in general, and this work is providing enough theoretical analysis for this challenging problem. In particular, a novel statistical test is proposed as a subroutine of the main algorithm. However, I have a few concerns : 1) When the mean of the rewards are highly variable, but the best arm is fixed, the bound in Theorem 2.4 will be linear in T. 2) Though I appreciate the novelty of statistical tests, both online and offline versions require an exhaustive search (finding two indices n_1 and n_2). 3) The main challenge in making an adaptive algorithm is distinguishing non-stationarity. Once that is done by tests, Can't we combine that with algorithms like Exp3 or other adversarial methods? I think in order to have more impactful results, the bounds should be tighter to recover static or non-adaptive scenarios up to log factors. A minor latex issue : footnote 4 in the line 242 appears in the next page (page 7) About the references: [13] and [14] are published in ICML and AISTATS, respectively.

Confidence in this Review

2-Confident (read it all; understood it all reasonably well)


Reviewer 4

Summary

The paper presents an algorithm that has sublinear dynanmic regret in the adversarial setting under limited variation in the rewards, without apriori knowledge of the magnitude of this variation. The algorithm has separate exploration and exploitation stages, and a pair of statistical tests of stationarity is proposed for each, such that jointly the dynamic regret is sublinear when the adveraries variation grows sublinearly

Qualitative Assessment

Sublinear regret against arbitrary sequences is a naturally interesting benchmark in the adversarial setting, and the variation of the rewards a sensible restriction to make it feasable. By removing the need for a priori knowledge of the variation it moves the bandits with sublinear dynamic regret closer to practical. The result is interesting if highly theoretical at this point. I would have valued the paper a lot more if some discussion of empirical finite sample performance of the proposed algorithm had been explored, even if such results where dismal. While there is sufficient value in pure theory and this work clearly represents a very substantial effort in that front, the total lack of exploration of practical aspects (beyond a cursory mention of one of the tests being polynomial time) makes it much harder to see it having a impact outside theory in the near future.

Confidence in this Review

1-Less confident (might not have understood significant parts)


Reviewer 5

Summary

The paper studies the stochastic Multi Armed Bandit problem, where the distributions of the arms could change over time. The authors approach the problem by first proposing a testing procedure to detect if there is a significant change in the means in an offline set (Test 1) and online series (Test 2) of independent random variables. They use these tests to design an online alg for the problem in the case of two arms The alg could be outlined as follows: - It dynamically divides the time horizon into blocks; each block consists of an exploration phase, and a possibly null exploitation phase. - During the exploration phase, the alg randomly pulls each arm with equal probability. If a change in either of the arms is detected, it goes to the next block (nil exploitation phase). If no (essential) change is detected and one arm is "significantly" better than the others, it goes to the exploitation phase. - The exploitation phase favors playing the better arm, but the length of the phase and the probability of playing the perceived better arm are weighed according to the probability that the exploration phase makes the correct identification. The main novelty of the algorithm is the proposal of Test 2, which requires the decision maker to actively and explicitly detect the change in the arms' distributions. This appears a novel aspect (at least to my knowledge) that is not featured in the current stochastic and adversarial MAB literature. While the paper is very closely related to a recent work by Gur, Zeevi and Besbes (2014), a main difference is that the amount of variation in the arms distributions is not known in the current paper's setting.

Qualitative Assessment

Minor typos: Line 238 - 239: In the definition of \epsilon_2(n_1, n_2) on page 6 (between line 238 and 239), the quantity is undefined when n_2 = n, n_1 = 1. Is there a typo? Line 302: \Delta \geq \tau^{-\lambda^2} should be \Delta \geq \Theta(\tau^{-\lambda^2}) Line 424: The reference for Hoeffding inequality is missing. Suggested additional references: A. Garivier and E. Moulines, "On upper-confidence bound policies for non-stationary bandit problems" in Algorithm Learning Theory (ALT), ser. Lectures notes in Computer Science, vol. 6925, 2011 Comments: The proposed algorithm is novel in the sense that it actively detects if there is any change in the arms' distribution, which is quite different from the current literature, including the one above. Designing online algorithm that competes well against the dynamic benchmark is clearly an important topic in MAB, and the paper makes a major step by designing such an algorithm without knowing the variation budget. Indeed, previous works such as [1, 7] referred in the manuscript requires knowing the variation budget beforehand to tune the learning rate, and it is quite surprising that one can achieves a sublinear regret without the knowledge. Thus, I think this submission will be interesting to the MAB community. The paper is quite clearly written, given the level of technicality. Finally, what is the regret order bound with K arms?

Confidence in this Review

2-Confident (read it all; understood it all reasonably well)


Reviewer 6

Summary

The paper tackles the adversarial bandit problem while competing with an optimal sequence of actions. The authors propose an algorithm with a regret bound scaling with a variation parameter. Their algorithm do not require any prior knowledge about the variation. They also propose an analyse a statistical test used to identify non-stationary within a sequence on independent random variables.

Qualitative Assessment

- The paper is well written and clear. - There are some weaknesses in the bibliography, switching-bandit algorithms with dependencies on the number of switches (EXP3.S is however cited[7]) and gaps between action's mean rewards being not described (Discounted UCB, Sliding Window UCB, EXP3 with reset). The detector of non-stationary has also some similarities with the SAO (Stochastic and Adversarial Optimal) algorithm that starts with a stationary algorithm and then switches to EXP3 if a non-stationarity is detected. - The proposed algorithm is well constructed and is a step-forward from the main reference [1]. While not having a major practical impact (has an high regret in stationary regime and in highly non-stationary regime, even if the best action does not change) it resolves and open problem and gives some closure the the analysis using the variation. - After the post rebuttal discussion, we agreed to rank this paper at poster level (5: 4->3).

Confidence in this Review

2-Confident (read it all; understood it all reasonably well)